# Sharks associated with a large sand shoal complex: Community insights from longline and acoustic telemetry surveys

**Eric Reyier**[1]*, **Bonnie Ahr**[1], **Joseph Iafrate**[2], **Douglas Scheidt**[1], **Russell Lowers**[1], **Stephanie Watwood**[2], **Brenton Back**[1]

1 Herndon Solutions Group, LLC, NASA Environmental and Medical Contract, Kennedy Space Center, Merritt Island, Florida, United States of America, 2 Naval Undersea Warfare Center Division Newport, Newport, Rhode Island, United States of America

* eric.a.reyier@nasa.gov

**Data Availability Statement:** All relevant data are within the paper and its Supporting Information files. At the request of study collaborators, details of acoustically tagged animals detected locally are

## Abstract

Offshore sand shoals are a coveted sand source for coastal restoration projects and as sites for wind energy development. Shoals often support unique fish assemblages but their habitat value to sharks is largely unknown due to the high mobility of most species in the open ocean. This study pairs multi-year longline and acoustic telemetry surveys to reveal depth-related and seasonal patterns in a shark community associated with the largest sand shoal complex in east Florida, USA. Monthly longline sampling from 2012–2017 yielded 2,595 sharks from 16 species with Atlantic sharpnose (*Rhizoprionodon terraenovae*), blacknose (*Carcharhinus acronotus*), and blacktip (*C. limbatus*) sharks being the most abundant species. A contemporaneous acoustic telemetry array detected 567 sharks from 16 species (14 in common with longlines) tagged locally and by researchers elsewhere along the US East Coast and Bahamas. PERMANOVA modeling of both datasets indicate that the shark species assemblage differed more across seasons than water depth although both factors were important. Moreover, the shark assemblage detected at an active sand dredge site was similar to that at nearby undisturbed sites. Water temperature, water clarity, and distance from shore were habitat factors that most strongly correlated to community composition. Both sampling approaches documented similar single-species and community trends but longlines underestimated the shark nursery value of the region while telemetry-based community assessments are inherently biased by the number of species under active study. Overall, this study confirms that sharks can be an important component of sand shoal fish communities but suggests that deeper water immediately adjacent to shoals (as opposed to shallow shoal ridges) is more valuable to some species. Potential impacts to these nearby habitats should be considered when planning for sand extraction and offshore wind infrastructure.

being withheld including identification numbers, size, sex, and release locations.

**Funding:** This work was funded by the US Bureau of Ocean Energy Management (BOEM) Marine Minerals Program via Interagency Agreement M13PG00031 with the Naval Undersea Warfare Center, as well as by the National Aeronautics and Space Administration (NASA) via the NASA Environmental and Medical Contract #80KSC020D0023. Both sponsors provided general guidance as to information needs, approved the general study design, and reviewed the manuscript prior to submission. Data collection and analyses, interpretation of results, and decision to submit for publication was made solely by study authors. In addition, species-level information presented in acoustic telemetry analyses was made possible by direct funding to tagging agencies that contributed these data (see acknowledgements).

**Competing interests:** The authors have declared that no competing interests exist.

## Introduction

As a group, sharks have proven highly prone to population declines from fishing harvest and habitat loss, and increased management intervention is urgently required in many regions of the world [1–3]. Understanding the habitat needs of sharks is a central component of effective management policy, especially for coastal species that are most directly exposed to human disturbances [4]. Habitat considerations are already incorporated into management policy in some countries. In US waters for example, where there is evidence of recent recovery in shark abundance and diversity [5,6], most species now have recognized Essential Fish Habitat (EFH) whose boundaries are established to minimize habitat impacts from commercial fishing, dredging, military operations, and other human activity [7]. Yet even here, shark habitat requirements and seasonal distributions are often only broadly described, as is the function and relative value of different marine habitat types in supporting shark populations and communities.

Several fishery-independent shark surveys are underway in continental shelf waters of the US Atlantic and Gulf of Mexico (see Peterson et al. [6] for a review). These efforts [5,8–11], based on traditional sampling gears including longlines and gill nets, are designed, in part, to better describe shark habitat needs and assemblage structure. Because these surveys often span wide stretches of coastline, replication at any given location is low with shark abundance and diversity insights largely focusing on the role of oceanographic factors (e.g., temperature, chlorophyll, depth, salinity) [12]. Year-round sampling of sharks associated with specific geomorphological features of the continental shelf are uncommon but can provide details on seasonal abundance trends and habitat associations that are often lacking in coarse-scale regional surveys. Such surveys are also easier to complement with acoustic tagging that reveals patterns in space use, improves the detectability of rare and rapidly migrating species, and overcomes certain known biases associated with traditional sampling gears.

Offshore sand shoals are submerged ridges, bars, or banks formed from sand or gravel that are shallower in depth than surrounding areas [13]. In addition to their complex bathymetry, shoals are generally characterized by higher wave energy, elevated turbidity, and heterogeneous sediments and ocean currents when compared to adjacent deeper water [14]. There is growing interest in understanding the habitat value and species composition of shoal communities because these features are important sand sources for coastal restoration projects. Sand is removed from offshore shoals with large dredges and deposited along the shoreline to counteract erosion resulting from storms, human development, and rising sea levels. Demand for sand from these offshore deposits is expected to grow rapidly in the US and around the world [15]. Shoals are also preferred sites for offshore wind farms since their shallow depth and soft sediments reduce turbine construction and maintenance costs [16].

Shoal fish communities have received less attention than those associated with other shelf habitats but they appear to be unique, albeit sometimes depauperate, species assemblages [17–21]. Dredging can negatively impact fish communities through direct mortality of immobile species in the dredge, removal of benthic invertebrate prey, as well as temporary increases in noise and turbidity which may result in avoidance responses [22,23]. Previous shoal fish surveys have relied heavily on camera sleds and trawls, with catches dominated by small species. Although shoals are common shelf features in many regions (and cover five percent of the shelf by area in the US South Atlantic [12]), sharks have never been a focus of shoal fish surveys, presumably due to their high mobility and naturally lower density on the open shelf. Nonetheless, most coastal shark species depend on the benthos to some degree for foraging, predator refuge, and navigation and therefore may also be impacted by shoal dredging activity and other human disturbances.

The overarching purpose of this study is to document the shark community associated with a large sand shoal complex in east Florida, USA, a region of high shark diversity—and intensive research—that serves as an overwintering area for many species. Our specific goals are to explicitly test whether the shark assemblage differs between shallow shoals and adjacent deeper water, as well as across seasons, and to test whether the community observed at an active dredge site differs from that of a nearby undisturbed site. Moreover, we then explore how potentially influential habitat factors such as water depth, temperature, turbidity, and sediments, help explain patterns in the shark community. Notably, this study is strengthened by complementing traditional longline sampling with passive acoustic telemetry over periods of five and six years, respectively. Longlines target a broad assortment of coastal sharks, providing estimates of abundance, size, and sex, while acoustic telemetry allows for detailed monitoring of shark habitat associations for species implanted with acoustic transmitters. Results are intended to improve our understanding of shark community patterns in the US southeast while helping refine best practices for sand extraction activities that minimize habitat impacts to this important group of fish.

## Materials and methods

### Study area

Cape Canaveral was selected as the study area due to the presence of the most expansive sand shoals on the Florida east coast and because it typifies other cape-associated shoals in the US southeast. Prominent features include the Southeast Shoal and Chester Shoal, shore-connected shoals with minimum depths of 2 and 4 meters (m), respectively, as well as several smaller detached shoals located farther offshore (Fig 1). The shallow wave-swept shoal ridges are generally comprised of medium to coarse quartzose-mollusk sand and shell. The deeper troughs between or in the lee of the shoals retain finer sand and muds with the most significant deposits found in the Canaveral Bight [24,25]. The study area includes one active 5 km$^2$ dredge site that has served as a sand source for eight beach nourishment projects since 2000. Hard-bottom substrates are sparse in the core project area but natural limestone reef outcroppings are common a few kilometers to the east and north of the shoals. The region exhibits a strong north-south gradient in winter water temperature due to the divergence of the warm Florida Current from the coastline [26]. No major coastal inlets occur locally so salinity remains roughly 36 psu year-round. Human development is limited to widely spaced NASA and US Space Force rocket launch infrastructure although commercial and recreational shark fisheries occur within the project footprint.

### Longline sampling

A 427 km$^2$ longline sampling universe was established to encompass the major shoal features off Cape Canaveral (Fig 1). Sixteen bottom longline sets were conducted monthly for five years from October 2012–September 2017 with samples divided equally among a shallow and deep depth zone. The shallow zone included shoal ridges less than 6.1 m (20 ft) depth, while the deep zone included deeper shoal flanks and the troughs between shoals out a maximum depth of 20 m (66 ft). Within each zone, new sample points were selected each month using a random point generator in ArcGIS 10.3 (ESRI, Redlands, CA). Additional targeted (i.e., non-random) longline sets were performed periodically, primarily in winter in the Canaveral Bight, to obtain additional sharks for acoustic tagging. Catch from these sets is presented separately and excluded from statistical analyses.

Each longline set spanned 617 m with a mainline of 318 kg monofilament anchored to the sea floor and marked at each end with large floats. Forty gangions were attached to each set,

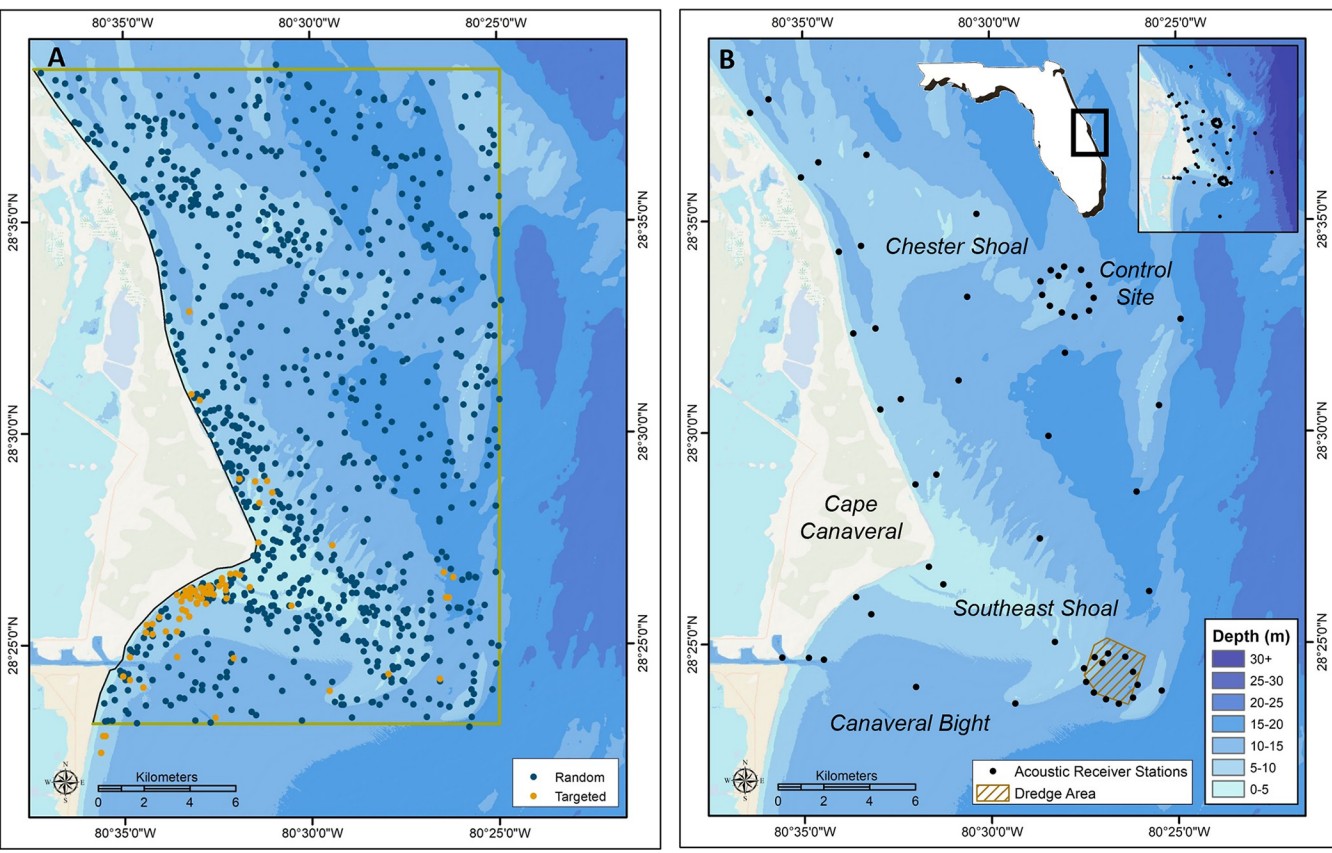

**Fig 1. Canaveral shoals study area.** (A) Locations of all longline sets 2012–2017. (B) Map of acoustic receiver stations 2013–2020 as well as the active sand dredge site and nearby control site. Map B inset includes locations of five reef receiver stations established farther offshore.

spaced at 15.2 m intervals. Gangions consisted of a stainless steel clip, 0.7 m of 181 kg test monofilament, with 12/0 and 15/0 circle hooks alternating along the mainline. Bait consisted exclusively of mullet (*Mugil* spp.), a regionally abundant forage fish. Longline soak time averaged 48 (± 24) minutes from the midpoint of deployment to midpoint of retrieval, with the mainline deployed and retrieved using a small electric winch. The short gangions, short soak times, and two hook sizes were designed to target a broad assortment of managed fishes (e.g., sharks, red drum [*Sciaenops ocellatus*], reef fish), increase sample replication, and reduce the likelihood of attracting fish from alternate habitats. Sampling occurred during daylight from an 8-m skiff embarking out of Port Canaveral. Habitat data collected at each set location included water temperature (˚C), dissolved oxygen (mg/l), salinity (psu), all measured in the middle of the water column (mean 2.9 m and 6.2 m for readings at shallow and deep sites, respectively) with a YSI 600XLM water quality sonde calibrated daily. Secchi depth (i.e., water clarity), mean water depth (averaged from boat depth sounder readings at the origin and end of each set), seafloor slope (the depth difference between origin and end of each set), distance from shore (km), and GPS coordinates were also recorded. A summary of longline habitat conditions under which each species was captured is presented in S1 Table.

All captured sharks were identified, sexed, and measured to the nearest 0.5 cm precaudal length (PCL), fork length (FL), and natural total length (TL). Individuals were classified as either neonate/young-of-the-year (YOY), juvenile, or adult through examination of claspers in males, the degree of umbilical scar healing in young animals (a sign of recent birth), and

published lengths of 50% maturity. All animal capture, handling, and tagging was conducted in accordance with a State of Florida Special Activity License SAL-12-512-SR and renewals, National Marine Fisheries Service Biological Opinion F/SER/2011/05647, and under the auspices of Kennedy Space Center Institutional Animal Care & Use Committee protocol GRD-06-049. Further details on longline and acoustic telemetry survey methods are found in Iafrate et al. [27].

## Acoustic telemetry

An existing acoustic telemetry array at Cape Canaveral (hereafter the 'Canaveral Array') was utilized from December 2013 to February 2020 (6.25 years) to independently assess the distribution of acoustically tagged fishes in the project area (Fig 1). The Canaveral Array is located near the center of the FACT Network, a multi-agency collaboration tracking fish and sea turtle movements in the US south Atlantic and Bahamas [28]. Fifty-seven acoustic receiver stations (Vemco VR2W and VR2AR, Innovasea, Nova Scotia) were permanently deployed on and immediately adjacent to the shoals from the surf zone out to 14 km offshore (2–16 m depth). Analogous to longline sampling, each station was deployed in the shallow (<6.1 m, n = 27) or deep (>6.1 m, n = 30) depth zone. Most stations were arranged in a non-overlapping grid layout although the design also included a 12-station receiver ring surrounding the sand dredge area on the Southeast Shoal and an identical 'control' ring at a reference site 15 km to the north on Chester Shoal. This control site was similar to the dredge site in many respects including water depth, distance from shore, and sediment qualities. These rings were used to explore for any large-scale differences in fish use of the dredge site vs. nearby undisturbed sites. To avoid dredge strikes, dredge site stations were deployed on the site boundary or within setbacks around identified rocket debris, and thus monitored disturbed and undisturbed benthic habitat in equal proportions, while control site stations monitored only undisturbed habitat. In September 2015, five additional stations were established along the offshore reef tract (10–22 km offshore, 15–25 m depth) to document any animal movements between shoals and reefs. All 62 stations were anchored to the seafloor with large sand augers or 40 kg metal disk weights, and with receivers retrieved twice annually for downloads using SCUBA. Multi-week range testing trials at three Canaveral Array stations confirmed that receivers could detect high power acoustic tags with a 50% efficiency at roughly 400 m on average [27].

Five shark species were acoustically tagged including adult and large juvenile finetooth (*Carcharhinus isodon*; n = 61), blacknose (*C. acronotus*; n = 60), and Atlantic sharpnose sharks (n = 44), as well as juvenile scalloped hammerhead (*Sphyrna lewini*; n = 39) and lemon sharks (*Negaprion brevirostris*; n = 30; Table 1). Sharks were primarily captured by longlines but also by hook-line (sharpnose shark), cast net (lemon sharks), and gill net (scalloped hammerhead)

**Table 1. Details of sharks acoustically tagged in the Canaveral Shoals study area.**

| Target Species | No. | Mean FL (Range) | Sex Ratio (F:M) | Tagging Dates | Tag Model, Ping Interval (sec), Batt Life (yrs) |
|---|---|---|---|---|---|
| Finetooth Shark | 61 | 104 (64–130) | 39:22 | Dec 2013–Sep 2016 | V16–4H or –6H, 90, 5.2–9.7 |
| Blacknose Shark | 60 | 96 (89–115) | 33:27 | May 2014–Sep 2016 | V16–4H or –6H, 90, 5.2–9.7 |
| Sharpnose Shark[b] | 44 | 74 (67–83) | 24:19 | Jul 2016–Aug 2017 | V16–4H, 90, 5.2 |
| Lemon Shark[a] | 30 | 87 (65–143) | 15:15 | Dec 2009–Dec 2020 | V16–4H or –6H, 90, 5.2–9.7 |
| Scalloped Hammerhead[ac] | 40 | 45 (37–115) | 17:23 | May 2013–Sep 2014 | V13–1L, V16–4H, 90, 2.2–5.2 |

[a] Most lemon and hammerhead sharks were tagged prior to dedicated monitoring of Canaveral Shoals but were included in analyses if detected after December 2013.

[b] Considered *Rhizoprionodon terraenovae* (not *R. porosus)* based on local genetic findings [29].

[c] Considered *Sphyrna lewini* (not *S. gilberti*) based on local genetic findings [30].

at multiple locations on the shoals, and across years. All sharks were measured and transferred to a seawater tank where they were oriented ventral side up to induce tonic immobility, and a small pump was used to circulate tank water across their gills. After injection of 2–5 cc of lidocaine, a ~2 cm incision was made along the ventral midline, a coded acoustic transmitter was inserted, and the incision and was closed with absorbable sutures and Vetbond™ tissue adhesive. Sharks were allowed to recover in the tank briefly before release at their capture site.

Numerous other sharks tagged elsewhere along the US Atlantic coast and the Bahamas were also seasonally present in the Canaveral Array and were identified to species by consulting tag lists maintained by the FACT and Atlantic Cooperative Telemetry Networks. By convention, these animal detections belong to the tagging organization [28]. References to sharks released by other groups are included here with written permission in all instances and in a manner that does not reveal size, sex, or migration details that are the focus of other ongoing studies. All shark detections were then combined into a single database and screened with a custom R script to remove any non-moving tags (presumed mortality events) or false detections that occasionally arise from tag signal collisions.

As with longlines, various habitat variables were also paired to the acoustic telemetry dataset including water temperature, water clarity, depth, seafloor slope, distance from shore, and latitude. Temperature and water clarity are not continuously monitored so satellite-derived monthly values were obtained for each station using the jplMURSST41 sea surface temperature dataset (0.01˚ resolution, daily composite) and erdMH1kd4908day K490 irradiance dataset (i.e., the mean diffuse attenuation coefficient at wavelength 490 mm, a proxy for turbidity; 4 km resolution, 8-day composite). These data are archived on the NOAA CoastWatch ERDDAP [31] and were queried using the rerddapXtracto package in R [32]. Seafloor slope at each receiver was calculated as the maximum vertical relief within 500 m of a station based on published NOAA charts. Sediment percent fines (i.e., muds) and organic content estimates were also available for all stations from a concurrent study of benthic communities on the Canaveral Shoals [25]. Samples were drawn from surficial cores collected once at each station via SCUBA. Fines consisted of sediment grains <63 um diameter as determined by wet sieving through a series of US standard sieves. Organics were the materials lost after ashing dried sediment samples in a muffle furnace at 500˚C for four hours. A summary of habitat conditions under which all species were detected is found in S2 Table.

## Data analysis

Longline catch was used to test for differences in overall shark abundance across water depths and seasons. Catch was first converted to catch-per-unit-effort (CPUE) in the form of sharks caught per 100 hook-hours of soak time. Monthly-averaged shark CPUE was then compared across *Depth Zone* (shallow ridge vs. deeper troughs) and *Season* (winter = Dec–Feb, spring = Mar–May, summer = Jun–Aug, fall = Sep–Nov) using a two-way ANOVA with an interaction term ($\alpha = 0.05$), and with least-square means to identify any pairwise differences across seasons. ANOVAs were run both with and without the numerically dominant Atlantic sharpnose shark included in CPUE values.

Patterns in the shark community were analyzed separately but similarly for longline and telemetry datasets. For longlines, species CPUE values from single sets were averaged for each combination of *Depth Zone* (2), *Season* (4), and *Year* (5.25). These 42 composite samples were necessary since individual sets often yielded only 1–2 species. For acoustic telemetry data, the total count of uniquely tagged sharks within each species detected at each station (coded *a priori* as deep or shallow) were tallied for each season (247 samples). This approach allowed us to explore *Season* and *Depth* trends while retaining *Station* as a factor nested within *Depth*. Raw

shark counts for each species were then converted to proportions. While variation in acoustic receiver performance due to water depth and other site-specific factors may bias the raw number of sharks detected at a given station, it should not affect species ratios. Notably, *Study Year* was not considered as a factor in telemetry analyses because acoustic tagging of additional shark species occurred as the study progressed.

Community differences across *Depth Zone* and *Season* were visualized with non-metric multi-dimensional scaling (MDS) plots in the PRIMER v7 software package [33]. Data were square root-transformed to allow less common species help explain differences between samples, and a sample similarity matrix based on the Bray-Curtis similarity coefficient served as the basis for MDS plots. Permutational multivariate ANOVA (PERMANOVA) models were then run to formally test for differences in the shark assemblage. PERMA-NONA is well suited for multi-species datasets containing a large fraction of zeros, and it allows for hierarchical designs, random effects, and tests for interactions [34]. For both longline and acoustic telemetry data, the PERMANOVA was a two-way crossed design (Type III sum of squares) with *Depth Zone* and *Season* as fixed factors plus an interaction term. The telemetry PERMANOVA also included a random effect of *Station* nested within *Depth Zone*. Data from all five reef stations and seven of 12 stations in both the dredge and control rings were excluded from the telemetry model to ensure that all included stations had identical deployment timelines and non-overlapping detection ranges. *Post hoc* homogeneity of multivariate dispersion (PERMDISP) tests were used to determine if observed community differences across depths and seasons were due solely to differing species complements or were also influenced by uneven sample variance within treatment groups. Two-way similarity percentages analyses (SIMPER [35]) were also run on the same longline and telemetry samples to identify which shark species were most responsible for community difference across depths and seasons.

Separately, a simple one-way PERMANOVA on telemetry data was used to test for assemblage differences between five more specific receiver station groups. These included the dredge site and control site, as well as all other nearshore shoal stations (<1.5 km from shore), offshore shoal stations (>1.5 from shore), and reef stations. The rationale for these latter three groups is that some sharks may closely follow the shoreline for navigation while others are known reef associates, behaviors which may also influence community patterns within the study area. Rarefaction curves were also generated to better assess overall species richness across these same station groups, a comparison otherwise complicated by the uneven receiver coverage. Longlines were not used to directly compare dredge and control sites because the small size of each site did not allow for sufficient sample replication.

Finally, the degree to which habitat conditions explained patterns in the shark community was explored with a BEST procedure, a routine that identifies combinations of measured environmental variables most strongly correlated to the species assemblage [36]. Certain oceanographic variables (e.g., temperature, water clarity) change rapidly across space and time so this routine was run using individual (not season-averaged) longline samples that contained more than one species, and on telemetry samples averaged for each combination of station and calendar month (not season). Longline habitat variables included water temperature, water clarity, depth (as a continuous variable), salinity, dissolved oxygen, seafloor slope, distance from shore, and latitude. Acoustic telemetry variables also included water temperature, clarity, depth, seafloor slope, distance from shore, and latitude, as well as sediment percent fines and percent organics. When necessary, habitat variables were log transformed to reduce skewness and normalized to remove the effect of differing measurement scales prior to running the BEST routine in PRIMER.

## Results

### Longline overview

A total of 978 longline sets (455 shallow, 445 deep, 78 targeted) were completed on or adjacent to the Canaveral Shoals from October 2012 through September 2017. Coastal sharks dominated samples, comprising 90% of all fish caught. Bony fish, primarily adult red drum, comprised 7% of catch, and benthic and pelagic rays represented 3%. In total, 2,595 sharks from 16 species were recorded (Table 2). Sharks were collected on 65% of sets with a maximum of four species occurring on any single set. Atlantic sharpnose shark alone accounted for 55% of shark captures, with blacknose (19%), blacktip (11%), and finetooth sharks (6%) also relatively common. For most species, catch was dominated by adults and large juveniles. All lemon sharks (n = 24) and bull sharks (n = 7) were juveniles but only 3% of sharks were classified as neonates including most spinner sharks and some scalloped hammerhead and sharpnose sharks. Notably, while catches of adult sharpnose shark were male-dominated, catches of blacknose, blacktip, and finetooth shark were female-dominated (Table 2). This female-skewed distribution

**Table 2. Numbers and sizes of sharks captured on longlines.** CPUE is expressed as fish per 100 hook hours and size as total length in cm. Targeted (i.e., non-random) longline sets are not included in Grand CPUE estimates.

| Species | Total | CPUE Shallow | CPUE Deep | CPUE Targeted | CPUE Grand | Mean Length (Range) | Female % | Adult / Juvenile / YOY % |
|---|---|---|---|---|---|---|---|---|
| Sharpnose Shark *Rhizoprionodon terraenovae* | 1436 | 2.73 | 8.06 | 0.00 | 5.37 | 85 (30–105) | 24.5 | 84 / 13 / 3 |
| Blacknose Shark *Carcharhinus acronotus* | 488 | 1.29 | 1.17 | 1.58 | 1.23 | 113 (73–155) | 66.3 | 73 / 27 / 0 |
| Blacktip Shark *Carcharhinus limbatus* | 277 | 0.87 | 0.43 | 0.99 | 0.65 | 132 (63–190) | 72.5 | 46 / 54 / 0 |
| Finetooth Shark *Carcharhinus isodon* | 157 | 0.45 | 0.15 | 0.80 | 0.30 | 132 (70–156) | 75.7 | 77 / 23 / 0 |
| Nurse Shark *Ginglymostoma cirratum* | 52 | 0.18 | 0.18 | 0.03 | 0.18 | 213 (135–278) | 36.8 | 56 / 44 / 0 |
| Bonnethead Shark *Sphyrna tiburo* | 40 | 0.09 | 0.09 | 0.21 | 0.09 | 105 (80–120) | 92.3 | 95 / 5 / 0 |
| Spinner Shark *Carcharhinus brevipinna* | 34 | 0.09 | 0.14 | 0.04 | 0.12 | 99 (64–210) | 47.1 | 18 / 6 / 77 |
| Scalloped Hammerhead *Sphyrna lewini* | 29 | 0.04 | 0.17 | 0.02 | 0.10 | 134 (47–200) | 22.2 | 11 / 82 / 7 |
| Lemon Shark *Negaprion brevirostris* | 24 | 0.12 | 0.02 | 0.01 | 0.07 | 149 (115–189) | 65.0 | 0 / 100 / 0 |
| Sandbar Shark *Carcharhinus plumbeus* | 22 | 0.04 | 0.10 | 0.03 | 0.07 | 168 (78–220) | 66.7 | 32 / 68 / 0 |
| Great Hammerhead *Sphyrna mokarran* | 13 | 0.05 | 0.03 | 0.03 | 0.04 | 265 (174–400) | 66.7 | 46 / 54 / 0 |
| Bull Shark *Carcharhinus leucas* | 7 | 0.02 | 0.01 | 0.00 | 0.01 | 188 (171–210) | 83.3 | 0 / 100 / 0 |
| Sand Tiger Shark *Carcharias taurus* | 5 | 0.02 | 0.01 | 0.05 | 0.02 | 183 (138–268) | 100.0 | 20 / 80 / 0 |
| Tiger Shark *Galeocerdo cuvier* | 5 | 0.01 | 0.03 | 0.00 | 0.02 | 191 (89–320) | 80.0 | 20 / 60 / 20 |
| Smooth Hammerhead *Sphyrna zygaena* | 1 | 0.01 | 0.00 | 0.00 | <0.01 | 152 | 100.0 | 0 / 100 / 0 |
| Dusky Shark *Carcharhinus obscurus* | 1 | 0.00 | 0.01 | 0.00 | <0.01 | 259 | 0.0 | 0 / 100 / 0 |
| Unknown Carcharhinid | 4 | 0.01 | 0.02 | 0.00 | 0.02 | 150 (100–200) | - | - |
| Total | 2595 | 6.02 | 10.62 | 3.79 | 8.30 | | | |

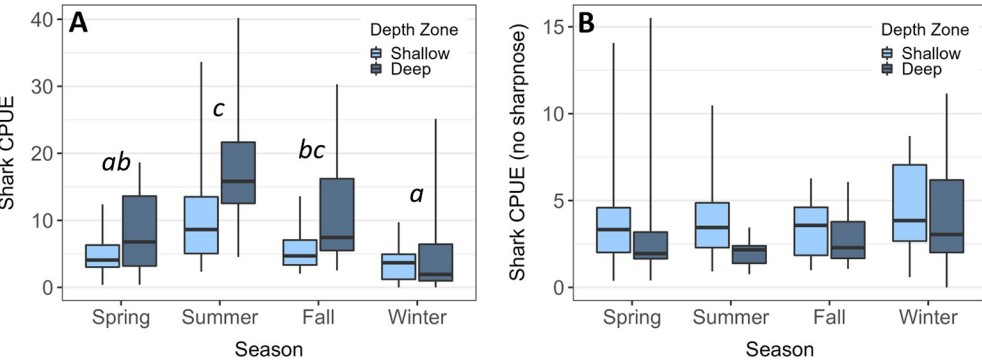

**Fig 2.** Boxplots of shark longline CPUE across water depth zones and seasons for (A) all species, and (B) excluding Atlantic sharpnose shark. Horizontal black lines represent the group median, boxes represent the interquartile range, and whiskers represent the full data range. Seasonal differences were only detected for the all-species ANOVA test with pairwise differences denoted with lowercase letters.

was most apparent from December through March, well before the typical spring pupping period, although gravid blacknose (n = 20), blacktip (n = 12), and finetooth (n = 26) were encountered over a wider timeframe from January to April or May each year.

Shark longline CPUE was significantly greater in the deeper troughs vs. shallow shoal ridges (two-way ANOVA, 10.6 vs. 6.0 sharks per 100 hook-hours; $F_{1,107} = 6.72$, $p = 0.011$; Fig 2). There was also a significant seasonal difference ($F_{3,107} = 15.28$, $p < 0.001$) with CPUE highest in summer (13.8), moderate in spring (6.6) and fall (8.8), and low in winter (4.0). These effects were due almost exclusively to the elevated abundance of sharpnose sharks in deeper water adjacent to the shoals, particularly in summer and fall (Fig 3). When sharpnose were excluded from this comparison, shark CPUE was modestly but significantly greater in shallow water (4.0 vs 3.1 sharks per 100 hook-hours; $p = 0.031$) but did not differ across seasons ($p = 0.387$). No depth by season interactions were detected. Besides sharpnose shark, other common warm season species included nurse and spinner sharks while cool-season species included finetooth, lemon, sandbar, and to some extent blacktip sharks (Fig 3). Catch of bonnethead and scalloped hammerheads suggested possible winter-spring abundance peaks while blacknose sharks were present in relatively even numbers year-round.

## Acoustic telemetry overview

A total of 567 acoustically tagged sharks across 16 species were detected within the Canaveral Array from December 2013 through February 2020. These included 219 sharks tagged on the Canaveral Shoals including blacknose (n = 57), finetooth (55), scalloped hammerhead (39), sharpnose (38), and lemon sharks (30), plus 348 sharks from twelve species tagged by researchers over a wide area from south Florida and the Bahamas to Nova Scotia, Canada (Table 3). Blacktip shark was the most commonly detected species (n = 114), and all species recorded in the Canaveral Array were also collected on longlines with the exception of white sharks (48) and common thresher shark (1). Acoustic telemetry better documented the presence of larger species relative to longline catches. Specifically, while sharks in the US small coastal management complex (e.g., sharpnose, blacknose, finetooth, bonnethead) represented 82% of the longline catch, they represented only 33% of sharks detected in the Canaveral Array, and the majority of these were tagged locally.

More individuals were detected in deep vs. shallow depth zones for all species except the lemon shark (Table 3), although this comparison is complicated by the superior detection

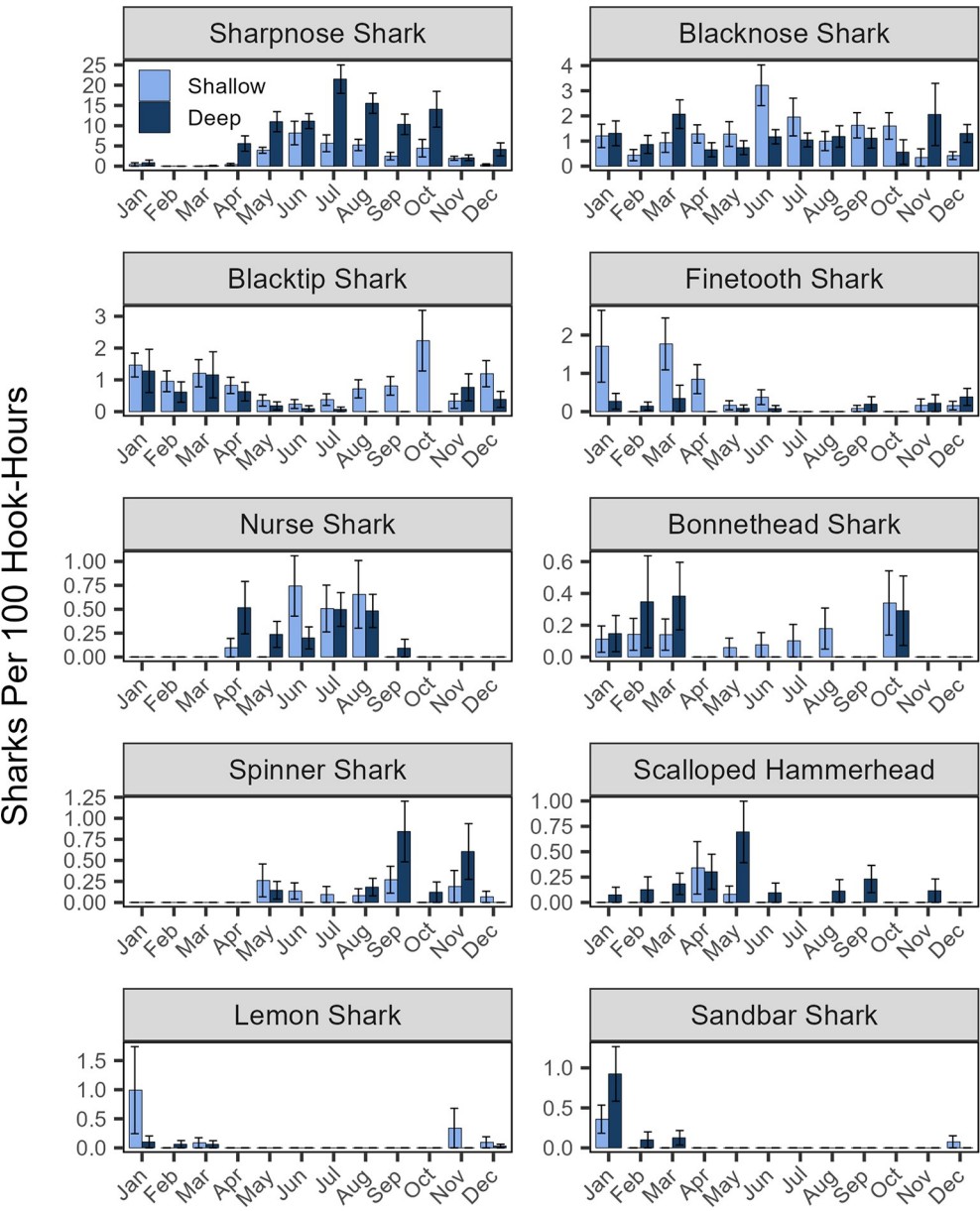

**Fig 3. Longline CPUE (+/- SE) by month and depth zone for ten most common shark species averaged across all five years of the study.** Note the differing y-axis scale for each species.

range of acoustic receivers in deeper water. While seven species were recorded in every month of the year, there were clear seasonal trends that closely mirrored longline catch trends for most species (Fig 4). Cool season sharks included blacktip, bonnethead, finetooth, lemon, sandbar, and white sharks, while warm season sharks included sharpnose, nurse and great hammerheads. Combined across years, winter was an especially productive season for acoustic telemetry monitoring with > 200 individual sharks detected each calendar month from November through April (peak 294 sharks in January) with a rapid decline in sharks detected from May-October (minimum 70 sharks in September).

**Table 3. Acoustically tagged sharks detected at Cape Canaveral December 2013–February 2020.** The number of acoustic receivers deployed in each habitat subset is listed in parentheses. Tagging organization designations are ordered based on number of individuals detected.

| Species | Acoustic Receiver Station Groups | | | | | |
|---|---|---|---|---|---|---|
| | All (62) | Deep (30) | Shallow (27) | Dredge (12) | Control (12) | Reef (5) |
| Blacktip Shark *Carcharhinus limbatus* [6,15,3,2,14] | 114 | 112 | 85 | 93 | 83 | 103 |
| Lemon Shark *Negaprion brevirostris* [2,1,3] | 73 | 58 | 64 | 35 | 27 | 28 |
| Blacknose Shark *Carcharhinus acronotus* [1] | 57 | 57 | 38 | 47 | 48 | 47 |
| Finetooth Shark *Carcharhinus isodon* [1] | 55 | 53 | 52 | 49 | 38 | 44 |
| White Shark *Carcharodon carcharias* [9,11] | 48 | 45 | 15 | 12 | 13 | 30 |
| Bonnethead Shark *Sphyrna tiburo* [15,16,7] | 39 | 38 | 19 | 11 | 18 | 30 |
| Scalloped Hammerhead *Sphyrna lewini* [1] | 39 | 27 | 21 | 10 | 8 | 3 |
| Bull Shark *Carcharhinus leucas* [2,3,12,14,4,8] | 38 | 35 | 20 | 20 | 22 | 26 |
| Sharpnose Shark *Rhizoprionodon terraenovae* [1] | 38 | 38 | 20 | 14 | 28 | 26 |
| Tiger Shark *Galeocerdo cuvier* [12,15,2,11] | 20 | 20 | 6 | 9 | 8 | 17 |
| Great Hammerhead *Sphyrna mokarran* [2,12] | 14 | 12 | 6 | 4 | 4 | 6 |
| Nurse Shark *Ginglymostoma cirratum* [2,8] | 10 | 10 | 2 | 3 | 4 | 8 |
| Sand Tiger Shark *Carcharias taurus* [5,9,10,17] | 9 | 9 | 5 | 2 | 2 | 3 |
| Sandbar Shark *Carcharhinus plumbeus* [3,10] | 7 | 7 | 1 | 2 | 4 | 4 |
| Spinner Shark *Carcharhinus brevipinna* [1] | 5 | 5 | 2 | 3 | 1 | - |
| Common Thresher Shark *Alopias vulpinus* [13] | 1 | 1 | 1 | - | 1 | 1 |
| Total | 567 | 527 | 357 | 314 | 309 | 376 |

[1]Current Project, [2]Bimini Biological Field Station, [3]Coastal Carolina Univ., [4]Cape Eleuthera Inst., [5]Delaware State Univ., [6]Florida Atlantic Univ., [7]Florida State Univ., [8]Harbor Branch Oceanographic Inst., [9]Massachusetts Div. of Marine Fisheries, [10]Monmouth Univ., [11]Ocearch, [12]Rosenstiel School of Marine and Atmospheric Science, [13]Stony Brook Univ., [14]Smithsonian Environmental Research Center, [15]South Carolina Dept. Natural Resources, [16]Univ. North Carolina Chapel Hill, [17]Univ. North Carolina Wilmington.

## Community differences across depths and seasons

Both longline and acoustic telemetry documented differences in the shark community through space and time (Fig 5). A PERMANOVA test of longline catch confirmed that *Season* explained over twice the variance as *Depth Zone* although both factors were significant ($p < 0.001$, S3 Table). Pairwise comparisons confirmed that all four seasons had unique shark assemblages with the greatest differences detected between summer and winter, whose samples were only 37% similar on average. A significant follow-up PERMDISP test for the factor *Season* ($p = 0.013$) suggests that community differences are due, in part, to uneven sample variation across seasons, not just differing species complements. Specifically, the shark catch in summer was more uniform than during other seasons due to the reliable presence of sharpnose sharks (Fig 5A).

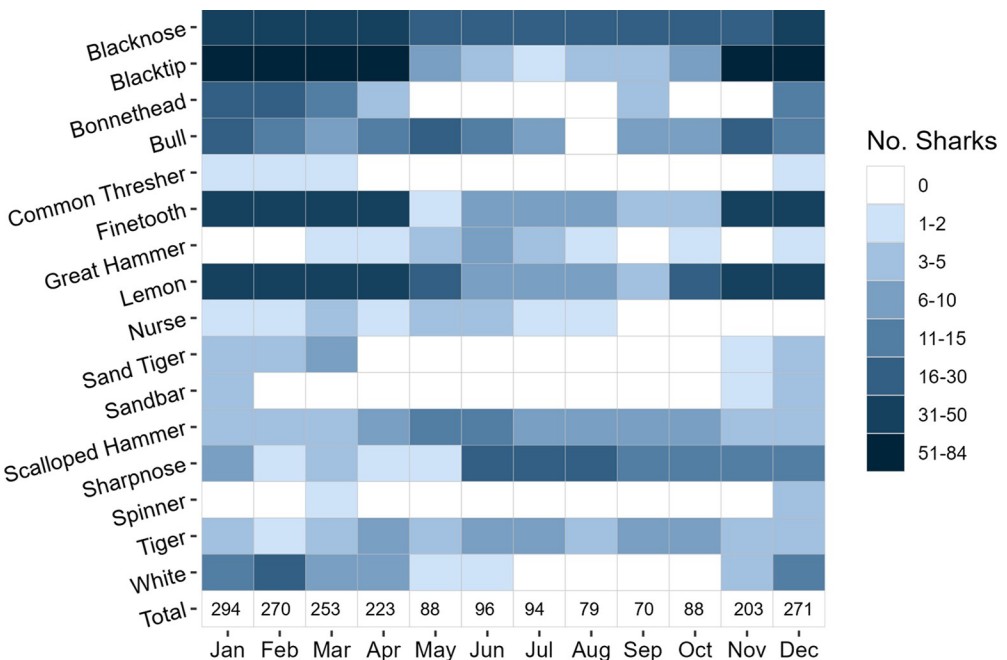

**Fig 4. Number of acoustically-tagged sharks detected in the Canaveral Array, December 2013 –February 2020.**
The total number of individuals detected in each calendar month is listed in the bottom row.

The shark community documented by the Canaveral Array also differed more across seasons than depth zones, with *Season* explaining nearly five times the variance as *Depth Zone* but both were again significant (PERMANOVA, $p < 0.001$, S4 Table). Interpretation is complicated by a significant season-by-depth interaction although examination of the MDS plot shows that the direction of the effects to be rather consistent across both factors (Fig 5B). The random effect of *Station* was also important ($p < 0.001$), explaining about as much variability as *Depth Zone* and demonstrating that differences in the shark community were detectable across individual receiver stations. As with longlines, the shark assemblage was distinct for each season, and with summer and winter samples again the most divergent (only 44% similar on average). PERMDISP tests again found that sample variability was unequal across seasons

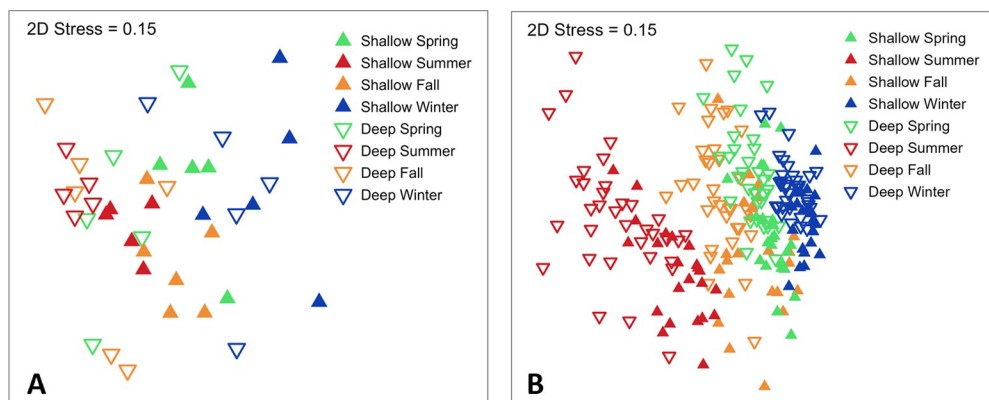

**Fig 5.** MDS plots of the shark community observed in (A) longline samples and (B) the Canaveral Array. Points represent the shark community for every combination of depth zone and season, with points closer together having more similar communities. Replicates within each longline group are drawn from each of the five years of the study while those in the Canaveral Array samples are drawn once from each acoustic receiver station.

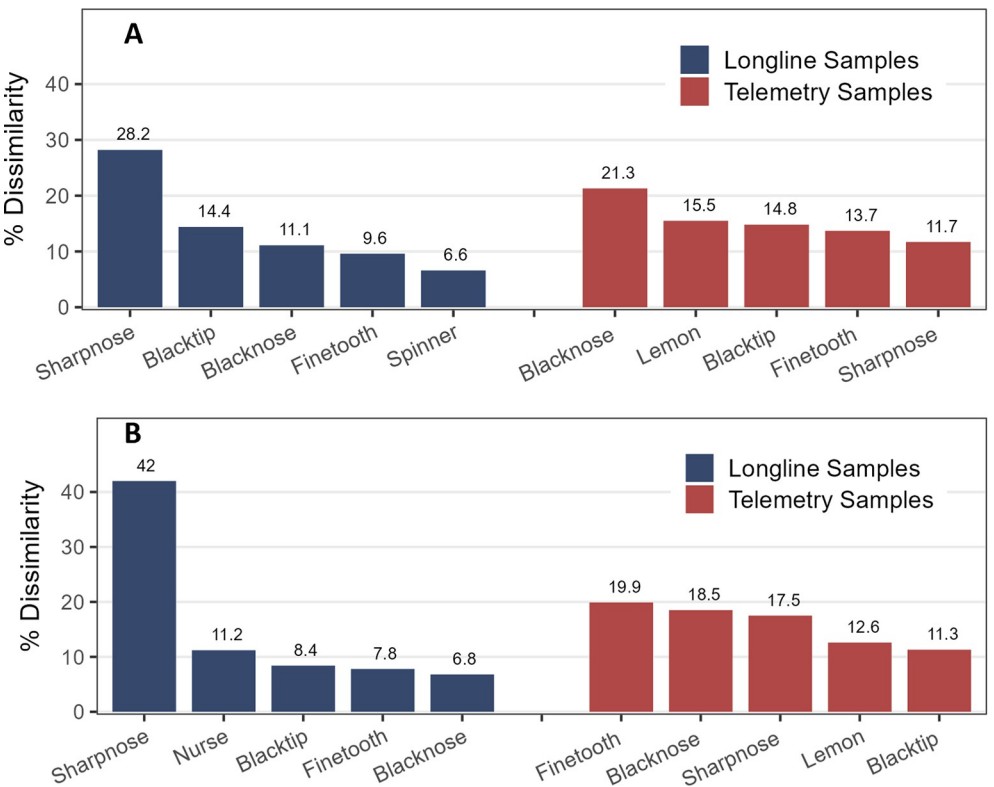

**Fig 6.** A SIMPER routine to determine which species contribute most to the dissimilarity (%) in the shark community across (A) Depth Zones and (B) Seasons for longline and telemetry datasets.

but not depths, but in contrast to longline catches, the species assemblage in the Canaveral Array was decidedly more uniform during winter than in other seasons.

Results of the longline SIMPER routine found that sharpnose shark alone, with its preference for deeper and warmer water, accounted for 28% of the dissimilarity in shark community across *Depth Zone* and 42% across *Season* (Fig 6A and 6B, respectively). Blacktip and blacknose sharks contributed moderately to community differences across depths (11–14%) while the nurse shark also helped discriminate across seasons (11%). In the telemetry dataset, blacknose sharks contributed most to differences across depths (21%) while finetooth, blacknose, and sharpnose sharks appeared similarly important (18–20%) in discriminating across seasons.

## Shark use of dredge and control sites, nearshore waters, and reefs

Combined across the six-year study, a similar number of shark species (15 vs. 16) and individuals (314 vs. 309) were detected by acoustic receivers at the sand dredge site vs. nearby control site (Table 3). Rarefaction curves also suggest that after standardizing for unequal receiver coverage, overall species richness at the dredge and control sites were similar to that of other offshore shoal sites, nearshore shoal sites, and reef sites (Fig 7A). Nonetheless, a PERMANOVA test and follow-on pairwise comparisons confirmed subtle but significant community differences between the dredge and control site as well as other receiver groups ($p < 0.001$; Fig 7B, S5 Table). The greatest contrasts were observed between nearshore shoal vs. offshore shoal stations and nearshore shoal vs. reef stations (samples 74% similar when averaged across the study), confirming that the shark community changes with increasing distance from shore. This transition was not dramatic, however, and numerous tagged finetooth, blacknose, blacktip, and lemon

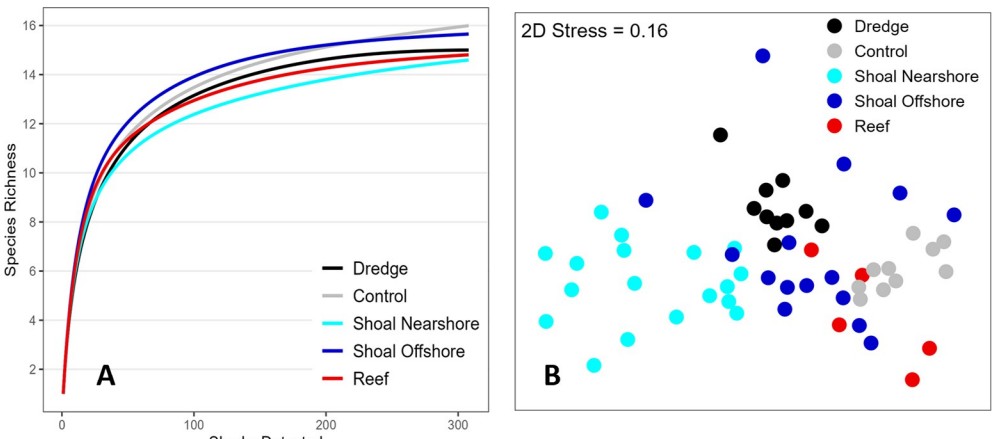

**Fig 7. Shark assemblage comparisons across dredge and control sites, other nearshore and offshore shoal sites, and reefs.** (A) Rarefaction curves comparing relative species richness after accounting for uneven numbers of acoustic receivers, and (B) MDS plot of the shark community detected at each receiver station in the Canaveral Array averaged across the study.

sharks, amongst the most common species in shallow longline samples, were routinely detected on the offshore reef tract well beyond the bounds of the Canaveral Shoals.

### Habitat factors influencing the shark community

Longlines and acoustic telemetry documented similar habitat associations for most shark species in the project area. In both datasets, for example, the lemon shark was encountered in shallower water (mean 5.4–6.7 m) and closer to shore (mean 1.2–1.5 km) on average than any other species. Finetooth sharks also showed similar preferences for shallow nearshore habitat across datasets while blacknose and blacktip sharks had intermediate depth and distance preferences. Conversely, sharpnose sharks exhibited a consistent preference for deeper water farther offshore (mean 10.3–11.9 m deep, 6.5–7.6 km from shore), as did sandbar, tiger, and great hammerhead sharks. The mean and range of habitat conditions for which each species as observed in longline samples and the Canaveral Array are provided in S1 and S2 Tables.

At the community level, the BEST procedure identified water temperature and water clarity as the habitat conditions most strongly correlated with the shark community in both longline and telemetry samples (Fig 8). The high rank of these two factors across both datasets reinforces their importance to shark distribution in the project area. Distance from shore and water depth also explained some community variation in both datasets. For longlines, the combination of water clarity (i.e., Secchi depth), water temperature, and distance from shore had the greatest overall correlation (BEST, $\rho_s = 0.280$). For telemetry samples, a combination of just sea surface temperature and water clarity (i.e., K490 irradiance), both obtained from satellite-based measurements, produced an even stronger correlation ($\rho_s = 0.510$). Notably, seafloor slope, a quality that helps distinguish sand shoals from adjacent shelf habitats, appeared uninformative ($\rho_s = 0.02$) in both datasets. Salinity and dissolved oxygen—available only for longline samples—and sediments % fines and % organics—available only for telemetry samples—also did not correlate with shark community structure.

### Discussion

This study was the first to directly sample a shark community associated with offshore sand shoals, dynamic but poorly surveyed features of continental shelves whose sand deposits are

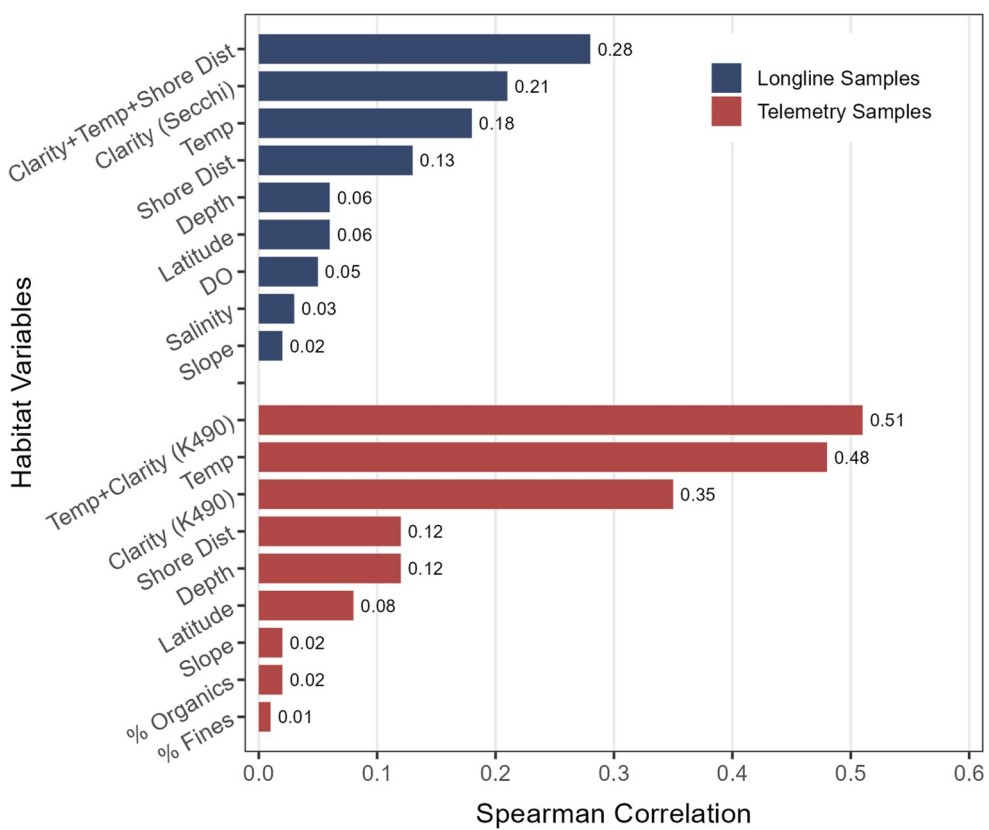

**Fig 8. A BEST procedure to determine which measured habitat variables correlate most strongly with the shark community.** Correlation values (Spearman ρs) for the top overall combination plus all individual variables are included.

valued for shoreline restoration projects and increasingly for renewable energy development. Our results confirmed that sharks as a group are abundant on shoals in east Florida and presumably analogous shoal sites in other regions. The opportunity to monitor spatial and seasonal changes in the local shark assemblage with both traditional longline gear and acoustic telemetry—perhaps the most unique aspect of this study—was made possible by working in a region of active shark research with a strong culture of researcher collaboration. Both approaches documented a similar suite of shark species (14 of 18 were held in common), improving confidence in the observed abundance and community trends, and providing a means to compare the strengths and limitations of each approach.

## Community differences across depths and seasons

Longlines and acoustic telemetry independently documented real but modest differences in the shark community between shallow shoal ridges and deeper water along shoal flanks and troughs. This finding was partly due to the rather narrow depth range sampled, out to a maximum of only 25 meters. Study objectives precluded sampling much deeper outer shelf waters where additional species would be encountered and community contrasts with shallow shoals would be more dramatic. Drymon et al. [10] for example, reported substantial changes in the shark assemblage across a 364-m depth gradient offshore Alabama with deep water samples producing nearly twice as many species. Distinct shark communities are also commonly associated with areas of extreme vertical relief such as seamounts and coral reef drop-offs [37,38].

Nonetheless, the present findings do appear to corroborate recent modeling by Pickens and Taylor [39] who found that proximity to sand shoals and seafloor slope are not strong abundance predictors for several shark species in shelf waters of the US South Atlantic and Gulf of Mexico.

All common species in this study were widely distributed in the project area suggesting that community differences across depths are due to each shark species using habitat on and adjacent to the Canaveral Shoals in different ways. For example, shallow water is often utilized by small sharks to reduce predation risk [40]. This strategy is adopted locally by young lemon sharks who aggregate each winter along the shoreline and inner shoals in water only 1–2 meters deep with only infrequent excursions farther offshore [41]. Finetooth shark, a small coastal species, also preferred shallower water than other species. In contrast, Atlantic sharpnose shark, the smallest species in the region, clearly selected for deeper water suggesting it has an alternative strategy to mitigate predation risk. Shallow shoals may also offer a temporary respite from adverse environmental conditions. Summer cold-water upwellings in east Florida often coincide with the formation of short-lived giant manta ray (*Manta birostris*) aggregations on shoal ridges, presumably to thermoregulate [27] although the behavior of sharks during these events is not well documented. In the northern Gulf of Mexico, certain shoals also serve as refugia for benthic fish and invertebrates when deeper water becomes hypoxic [42,43], events which also undoubtedly influence shark distribution.

Deeper troughs between and adjacent to shoal ridges may serve as high value foraging grounds for many sharks and other large marine predators. The prominent sand waves, invertebrate burrows, and fine sediments that are routinely scoured away by waves and strong currents on ridges, persist in deeper water, offering complex microhabitats that are preferred by many small benthic fishes and invertebrates [17]. Previous trawl and camera sled surveys suggest that deep water near shoals exhibit greater abundance of benthic fauna than shoal ridges themselves [19,20]. At Cape Canaveral, a trawl survey performed concurrently with the present study documented over double the fish density and biomass, and higher species richness in deeper troughs (10–18 m) than on ridges (<10 m) [44]. Moreover, space use analyses of several priority species tagged at Cape Canaveral found that blacknose and finetooth sharks, as well as benthic feeding red drum and loggerhead turtles, spent considerable time in deeper water between and adjacent to the shoals, and particularly in the Canaveral Bight, suggesting that these areas are important foraging grounds [27]. Even so, most coastal sharks also opportunistically feed on pelagic prey whose movements are not closely linked to water depth. Pelagic forage fishes, particularly the regionally abundant menhaden (*Brevoortia* spp.), are important prey for sharpnose, blacktip, spinner, and finetooth sharks [45], a preference which may explain why most sharks range widely through the project area including excursions to offshore reefs.

Changes in the shark assemblage across seasons were pronounced and somewhat anticipated. Long distance migrations, while only now being resolved in detail for many coastal sharks thanks to improved tagging technology (see [46–50]), have been broadly recognized for decades in the US Atlantic and many other regions of the world. Although not presented in detail here, a majority of blacknose, finetooth, sharpnose, and lemon sharks acoustically tagged at Cape Canaveral undertook northward spring migrations before returning to east Florida in fall [27,41], and over 200 sharks tagged by other researchers from South Florida and Bahamas to Canada migrated through the project area. These migrations result in a shark community that is in constant flux throughout the year. Thus far, seasonal variations in shark communities have been best described for estuaries (see [51–55]), habitats where annual variation in environmental conditions can be dramatic. Shark community surveys over the open continental shelf—especially those that include year-round sampling—are uncommon but similar patterns

should be expected and may be particularly stark at mid-latitudes where seasonal changes in water temperature and primary productivity are pronounced.

## Habitat influencing the shark community

Shifts in the shark community across depths and seasons are the culmination of sharks reacting to multiple cues in their local environment. Oceanographic conditions, prey abundance, competition, and predation risk (among other factors) all influence the distribution and behavior of each species. Of the measured environmental and habitat variables, changes in water temperature and water clarity were two factors that best matched changes in the shark community. As ectotherms, temperature directly influences shark metabolism and growth [56], embryonic development [57], and swimming speed [58], and mediates interspecies competition and predatory-prey relationships [59]. Consequently, water temperature is recognized as a primary factor dictating the geographic distribution, migrations, and community structure of sharks worldwide [60]. The role of temperature in structuring fish communities may be especially important east Florida where water temperature varies nearly 15°C across the year and where the shifting edge of the Florida Current can lead to rapid swings in coastal ocean temperatures and periodic upwellings [27].

Water clarity also influenced the shark assemblage in acoustic telemetry and particularly longline samples, where it appeared even more influential than temperature. Water over and adjacent to the Canaveral Shoals is often quite turbid due to waves breaking over the shallow seabed and resuspending fine sediments that accumulate in the lee of the shoals. Sharpnose shark strongly preferred clearer offshore water but other species, particularly finetooth, lemon, blacktip and blacknose sharks, were abundant in turbid conditions. Water clarity has also been shown to structure shark communities in other regions including Australia [61], South Africa [62], and the Gulf of Mexico [11]. Pelagic sharks often select for clear water [63] but several coastal species have been shown to select for high turbidity including sand tiger [49] and scalloped hammerheads [64]. Small fish and crustaceans often prefer elevated turbidity since it reduces individual risk of predation [65,66]. Foraging coastal sharks may move in concert with these prey species and thus spend more time foraging in turbid conditions; small sharks may themselves experience reduced predation risk in turbid water. Therefore, while there was no obvious association of sharks with shallow shoal ridges, these features may still serve an important function by introducing greater local variation in water clarity that are exploited by small coastal fishes and the sharks that depend on them.

Distance from shore was also moderately influential in structuring the local shark community. Common species including lemon, finetooth, and blacknose sharks were encountered relatively close to the beach in both datasets (1–3 km on average) while others such as sharpnose, nurse, and great hammerhead sharks were more abundant farther offshore (4–8 km on average). Notably, the visit duration of several sharks to stations in the Canaveral Array increases when near the beach [27] suggesting reduced swim speeds or more tortuous swimming paths, perhaps linked to foraging. These findings are consistent with a review by Sequeira et al. [67] who demonstrated that movement of marine vertebrates become more complex and less directed near the coast, regardless of taxonomy or body size, a convergence they link to increased microhabitat complexity in nearshore waters.

## Nursery function of shoals

Many coastal sharks rely on spatially discrete nursery grounds where females give birth and young spend their first months or years of life. Nurseries are defined as areas that support high densities of young sharks, sustain individuals for extended periods, and are used repeatedly

across years [68]. These sites are likely selected for the high productivity, reduced predation risk, and optimal temperature and salinity [68]. While some small coastal species including sharpnose and blacknose shark—the two most abundant species in longline samples—may forego the use of nurseries altogether [40], identifying nursery locations for sharks that require them is important because these sites often occur in shallow coastal waters that experience elevated human habitat degradation and fishing pressure.

Nearshore waters of east-central Florida have been recognized as a nursery for the scalloped hammerhead [69], spinner [70], and lemon shark [71]. Longline catches support these earlier assessments; all captured lemon sharks were immature, and most neonates were either spinner or scalloped hammerhead sharks. Nonetheless, anecdotal observations suggest that this gear was underestimating the density of young sharks. From 2012–2016 for example, YOY scalloped hammerheads (n = 369, mean FL = 39 cm) comprised 40% of all sharks collected in gill net sampling off Cape Canaveral in support of tagging and genetic studies ([30]; authors' unpublished data). In February 2017, nearly 1800 immature (~0.75–2.0 m TL) lemon sharks were video documented along a 29 km aerial survey along the Canaveral shoreline, roughly one shark every 16 meters (authors' unpublished data). None of these species strongly associated with offshore shoal ridges. Most YOY spinner sharks and hammerheads were collected in deeper water adjacent to the shoals while juvenile lemon sharks aggregate in the surf zone. Moreover, declining CPUE of finetooth, blacktip, and blacknose in March and April each year, and confirmed northward migrations of many tagged females at this time [27] suggest that pupping primarily occurs north of Cape Canaveral. Gill nets, which better target small sharks and avoid biases from bait type and hook size, are likely a more effective gear for shark nursery studies on sand shoals but can be more labor-intensive, reducing the replication needed to evaluate shark community patterns and habitat preferences. Regardless, this study did not document evidence that the shallow shoals themselves served a unique shark nursery function or that other species pupped nearby in especially large numbers.

## Implications for dredging

Sharks as a group are thought to face reduced direct risk from sand dredging compared to benthic rays and teleost fishes. As large and mobile predators, sharks are unlikely to suffer direct mortality from the dredge itself [72] and many coastal species have been shown to maintain large home ranges and exhibit low site fidelity, although reef-associated sharks are important exceptions. Sand dredge sites will represent a small fraction of an individual shark's overall activity space in most instances. Moreover, while many sharks forage on the seafloor, others feed primarily or opportunistically in the water column and may better tolerate disruption to benthic communities resulting from dredging operations. In this study, only modest differences between the shark assemblage were detected at the active dredge site versus the nearby control site. Moreover, behavioral assessments of the five shark species tagged at Cape Canaveral (detailed in [27]) demonstrated that individuals remained near a given receiver station for less than an hour on average, made regular inshore-offshore movements, and commonly undertook migrations spanning several hundred kilometers, all behaviors which minimize the effects of localized dredging disturbances. One major limitation of the present study is that the shark community comparison at the dredge vs. control site was purely descriptive and complicated by the repeated disturbance of the dredge site over the past 20 years. More rigorous study designs such as Before-After-Control-Impact (BACI) protocols are viable options when evaluating proposed sand borrow sites and would better reveal causal links between dredging to shark community structure.

Amongst the most significant impacts to sharks from dredging may occur beyond the boundaries of a dredge site itself. Dredging along shallow shoal ridges, for example, will increase water depth and can alter local wave field dynamics, sediment grain size, turbidity, and benthic communities in unpredictable ways [73,74]. The deeper troughs adjacent to and between shoals, particularly in the Canaveral Bight, were regularly utilized by most shark species. Exposing these areas of fine sediment to more powerful waves and ocean currents could degrade the quality of local shark nurseries and foraging grounds. Altered turbidity and sedimentation processes are also a recognized threat to reef habitat [75] and the sharks that rely on them. At Cape Canaveral, this risk is low because the main reef tract occurs several km east of the active dredge site, and sand is dredged from the leading edge and trough (not ridge) of the shoal. In areas where reefs lie closer to a dredge site, are dominated by sensitive corals, or where water clarity is naturally high, minimizing changes in turbidity and sedimentation during and after dredging should be a principal concern.

Offshore sand dredging is almost always accompanied by redeposition of the same sand along the shoreline to counteract beach erosion or rebuild marshes. These nourishment activities can themselves disrupt fish communities through burial and sedimentation of nearshore hardbottom [76]. Locally, lemon sharks may be the species of greatest risk from beach placement of sand. Young sharks gather each winter in semi-isolated runnels between the shoreface and offshore sandbar, in concentrations sufficient for east-central Florida to be classified as a Habitat Area of Particular Concern (one of only three shark species in the US to benefit from this designation). Beach renourishment activity, which also commonly occurs in winter to avoid impact on nesting sea turtles and shorebirds, would temporarily fill these runnels. The effect on lemon shark aggregations is hard to predict although the species experienced declines in survival after shoreline dredging operations in the Bahamas [77].

## Longline vs. acoustic telemetry comparison

Bottom longlines and acoustic telemetry separately documented similar single species and community trends in east Florida but each has limitations for surveying offshore sand shoals. Longlines are easily quantifiable and thus widely used for population assessments but with known biases related to shark size and behavior, hook size, and bait choice [78]. Bonnetheads, for example, feed primarily on blue crabs (*Callinectes* spp.) [79] and were likely underrepresented in the catch locally. Longlines will also miss rare species. While not a shark, acoustically tagged smalltooth sawfish (*Pristis pectinata*), now protected under the US Endangered Species Act, were regularly detected by the Canaveral Array in summer [80] but none were collected despite being prone to capture on baited hooks. Finally, longline sampling here was conducted during the day to maximize safety when working on the shoals. Studies have found that shoal fish assemblages can vary over a diel cycle [17], and day-night differences in catches rates have been reported for several coastal shark species [54,81].

Acoustic telemetry monitoring of shark communities are less influenced by these sampling biases but are only realistic in regions where multiple species are being tracked and researcher coordination is occurring. Due to varying management effort across nations as well as cost and logistical constraints, this approach remains unrealistic in many parts of the world. Moreover, acoustic telemetry cannot easily assess single species abundance or explore interannual community patterns since the number of tagged animals is in constant flux and represents an unknown fraction of the population. Finally, it is challenging to collect the needed environmental variables (e.g., temperature, turbidity, chlorophyll) at acoustic telemetry stations for which to rank habitat value although improvements in environmental data loggers and remote sensing capabilities are addressing this limitation.

## Conclusions

As a group, sharks are a prominent and year-round component of sand shoal fish community in east Florida and presumably in similar habitats in other regions. Nonetheless, most common species were widely distributed with relatively subtle differences in community composition observed between shallow shoal ridges and nearby deeper water, and across an inshore-offshore gradient, at least out to a depth of 25 meters. With their naturally high mobility and low site fidelity, sharks may be better able to avoid or overcome direct dredging disturbances, and no major community differences were observed between a previously dredged site and a nearby undisturbed site. Despite no strong affinities for shallow shoal ridges, shoals on the whole offer elevated habitat complexity when compared to much of the continental shelf. Coastal sand dredging operations and wind energy development projects, which already focus on minimizing disturbance to nearby reef habitat, should also consider impacts to soft-bottom substrates directly adjacent to shoals, and strive to preserve natural variation in seafloor bathymetry, sediments, and turbidity that likely sustain the most diverse shark community.

## Supporting information

**S1 Table. Habitat conditions under which each shark species was collected on longline sets.** Values are mean with range in parentheses. The mean and range across all longline sets is also provided.
(DOCX)

**S2 Table. Habitat conditions under which acoustically tagged sharks were detected on the Canaveral Shoals.** Detections from offshore reef stations are excluded. Values are mean with range in parentheses.
(DOCX)

**S3 Table. PERMANOVA test for shark community differences in the longline catch across seasons and depth zones.** Variance values estimate the amount of variation in the dataset explained by the factor while SD gives the square root of these values, and thus is in Bray–Curtis units. Pairwise tests of community similarity across seasons are also provided.
(DOCX)

**S4 Table. PERMANOVA test for shark community differences in Canaveral Array detections across seasons and depth zones.** Variance estimates the amount of variation in the dataset explained by the factor while SD gives the square root of these values, and thus is in Bray–Curtis units. Pairwise tests of shark community similarity across seasons are also provided.
(DOCX)

**S5 Table. PERMANOVA test for shark community differences in Canaveral Array detections across station groupings.** Groups include the offshore dredge site, offshore control site, all other offshore shoal stations (>1.5 km from shore), offshore reef stations, and nearshore shoal stations (< 1.5 km from shore).
(DOCX)

**S1 File.**
(XLSX)

## Acknowledgments

We thank Jen Bucatari, Deena Hansen, and Jake Levenson (BOEM), Jeff Collins and Lynne Phillips (NASA), and Jane Provancha (Herndon Solutions Group) for support with project

management. Chris Schumann, Karen Holloway-Adkins, Shanon Gann, Tim Kozusko, and Carla Bourtis assisted field data collections. Deb Murie (Univ. Florida) analyzed important sediment data from the Canaveral Shoals. Thanks goes to Jessica Greene and Laura Sparks from Naval Undersea Warfare Center Division, Newport for GIS support. Special thanks to Matt Ajemian and Mike McCallister (Harbor Branch Oceanographic Inst.), Jeremy Arnt, Caroline Collatos, Kelsey Spencer, and Dan Able (Coastal Carolina Univ.), Charles Bangley and Matt Ogburn (Smithsonian Environmental Research Center), Martin Benavides and Matt Kenworthy (Univ. North Carolina Chapel Hill), Beth Bowers (Florida Atlantic Univ.), Annabelle and Edd Brooks (Cape Eleuthera Institute), Keith Dunton (Monmouth Univ.), Dewayne Fox (Delaware State Univ.), Brian Franks (Jacksonville Univ.), Brian Frazier (South Carolina Dept. of Natural Resources), Mike Frisk (Stonybrook Univ.), Neil Hammerschlag (Univ. Miami), Danielle Haulsee (Stanford Univ.), Brian Keller (Univ. Arizona), Steve Kessel (Shedd Aquarium), Jeff Kneebone (New England Aquarium), Madeline Marens (Univ. North Carolina Wilmington), Greg Skomal (Massachusetts Div. of Marine Fisheries), and Matt Smukall (Bimini Biological Field Station) for allowing summaries of their acoustic telemetry data to be presented.

## Author Contributions

**Conceptualization:** Eric Reyier, Joseph Iafrate, Douglas Scheidt, Russell Lowers, Stephanie Watwood.

**Data curation:** Eric Reyier.

**Formal analysis:** Eric Reyier, Bonnie Ahr.

**Funding acquisition:** Eric Reyier, Joseph Iafrate, Stephanie Watwood.

**Investigation:** Eric Reyier, Bonnie Ahr, Douglas Scheidt, Russell Lowers, Stephanie Watwood, Brenton Back.

**Methodology:** Eric Reyier, Bonnie Ahr, Joseph Iafrate, Douglas Scheidt, Russell Lowers, Stephanie Watwood, Brenton Back.

**Project administration:** Eric Reyier, Bonnie Ahr, Joseph Iafrate, Stephanie Watwood.

**Resources:** Eric Reyier, Joseph Iafrate, Douglas Scheidt.

**Software:** Bonnie Ahr.

**Supervision:** Eric Reyier, Joseph Iafrate, Douglas Scheidt, Russell Lowers.

**Validation:** Bonnie Ahr.

**Visualization:** Bonnie Ahr, Joseph Iafrate.

**Writing – original draft:** Eric Reyier, Joseph Iafrate.

**Writing – review & editing:** Eric Reyier, Bonnie Ahr.

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
