## [Decision Letter · Decision Letter 0]

8 Mar 2023

PONE-D-23-02711Sharks associated with a large sand shoal complex: Community insights from longline and acoustic telemetry surveysPLOS ONE

Dear Dr. Reyier,

Thank you for submitting your manuscript to PLOS ONE. After careful consideration, we feel that it has merit but does not fully meet PLOS ONE’s publication criteria as it currently stands. Therefore, we invite you to submit a revised version of the manuscript that addresses the points raised during the review process.

This is a strong and compelling manuscript, but four areas need to be strengthened before it can be published:

I. Introduction:  Please re-phrase the goals of this study, to highlight that there is a hypothesis-driven component, with three objectives and an exploratory component.

For instance, the introduction could explicitly state that the paper integrates two approaches towards understanding how the overall shark assemblage differs.  The first hypothesis-driven approach addresses three patterns:

(1) spatially: comparing shallow shoals and adjacent deeper water

(2) temporally: comparing four seasons and

(3) compares the community at an active dredge site to a nearby control site.

The second exploratory component, seeks to identify other habitat factors that help structure the shark community, and integrates a variety of bathymetric and hydrographic datasets.  To strengthen this second component, I would ask the authors to explicitly list what patterns (or habitat variables) are being explored. 

Alternatively, if all of the environmental variables used to explore other habitat factors are already used in the hypothesis-driven approach, the paper could be focused on the first approach.  Currently, its unclear how the exploratory component of the paper enhances the hypothesis-driven component.  Are they really distinct?  Is this disparity based on the multivariate analyses: the exploration part uses NMS ordinations and the hypothesis-driven part uses ANOVAs and PERMANOVAS?

II. Methods:  Sometimes the explanation of the analytical techniques and associated decisions are not sufficiently explicit, but can easily be made more accessible in review (see line-by-line comments from one reviewer).  One major area where improvements are needed is to provide detailed explanations about the environmental data and the way they were collected:

Provide the depth range where the water column measurements were taken with the YSI sonde?  Since there are shallow (< 6.1 m) and deep (6.1 – 20 m) stations, did the depth of the measurements vary widely?  If so, can you please provide a summary of this information?   Provide the make and model of the YSI sonde and the resolution of the physical measurements. Moreover, please provide information about any validation of the data and about any blanks taken in the field or in the lab before/ after the field measurements.   For instance, using type-II water.Provide details about how the water depth and seafloor slope were collected.  Was the depth measured at the start and at the end of the set, and then the gradient was calculated as max depth minus min depth divided by the linear distance between them? Many variables are discussed, but there is no explanation of how they were collected, what is the resolution and precision / accuracy of the measurements, and how were these data merged with the acoustic and longline catch data.  Were these discrete data interpolated to assign values to the shark data?   I strongly suggest the authors provide a summary table, showing these details for all the environmental data.In particular, please provide more details about the temporal / spatial resolution of the remotely sensed data?  Also, please explain how were clouds dealt with?  Did you use or develop seasonal composites?  Did you use the maximum or mean values to do so?  It seems strange that the definition of the offshore domains is different for shoals and reefs:  offshore shoal stations (>1.5–13 km from shore) VERSUS offshore reef stations (10–22 km from the shore).  Why not do two offshore sections (shoal and reef) up to 13 km from shore and then consider a very offshore region (13 – 22 km from shore).  I am concerned that the current offshore stations are sampling different ocean habitats.  Can you show with your environmental data hat they do not differ in depth and water properties?Can you explain why a 2-D solution was selected for the MDS ordination?  Was a scree plot used?  Were the 2-D solutions the ones with the lowest stress? Or were 2-D solutions selected for the ease of visual interpretation?   Regarding the comparison of the dredge and the control site, you stated that: “Combined across the six-year study, a similar number of species (15 vs. 16) and individuals (314 vs. 309) were detected by acoustic receivers at the sand dredge site vs. nearby control site (Table 3).”  Could you generate rarefaction curves to compare the species richness and the diversity (and evenness) at these two sites?  This would be an easy way to visualize these patterns?   You could use the iNext R package.

III) Discussion:  A more focused and cohesive discussion, built upon the results, would strengthen the ms. 

The broad scope of the objectives sometimes results in parts of the discussion not being sufficiently explained (see reviewer comments). For instance, not enough attention is given to the comparison between the dredged and undisturbed site, despite this being one of the main aims of the manuscript. The manuscript would also benefit from a slightly more cohesive discussion, built upon the results. Some sections are not discussed in enough detail or are not well supported by the results, while others could be avoided (see detailed comments from reviewer). Namely, the role of the dredge site as a nursery is a key section in the discussion, but is not properly covered in the results.The discussion would also benefit from a stronger conclusion.  Perhaps this could be enhanced by streamlining the goals listed in the introduction and explicitly addressing these points.

IV) Finally, a few typos should be corrected.  Please read the paper carefully before re-submitting.

We look forward to receiving your revised manuscript.

Kind regards,

David Hyrenbach, Ph.D.

Academic Editor

PLOS ONE

Journal Requirements:

6. We note that Figure 1 in your submission contain map/satellite image which may be copyrighted. All PLOS content is published under the Creative Commons Attribution License (CC BY 4.0), which means that the manuscript, images, and Supporting Information files will be freely available online, and any third party is permitted to access, download, copy, distribute, and use these materials in any way, even commercially, with proper attribution. For these reasons, we cannot publish previously copyrighted maps or satellite images created using proprietary data, such as Google software (Google Maps, Street View, and Earth). For more information, see our copyright guidelines: http://journals.plos.org/plosone/s/licenses-and-copyright.

Reviewers' comments:

Reviewer's Responses to Questions

**Comments to the Author**

1. Is the manuscript technically sound, and do the data support the conclusions?

Reviewer #1: Partly

Reviewer #2: Yes

2. Has the statistical analysis been performed appropriately and rigorously? 

Reviewer #1: Yes

Reviewer #2: Yes

3. Have the authors made all data underlying the findings in their manuscript fully available?

Reviewer #1: Yes

Reviewer #2: Yes

4. Is the manuscript presented in an intelligible fashion and written in standard English?

Reviewer #1: Yes

Reviewer #2: Yes

5. Review Comments to the Author

Reviewer #1: This manuscript makes use of long-term longline catch data and passive acoustic telemetry to describe the shark assemblages of a sand bank system and investigate the temporal and spatial patterns in their distribution and abundance. The manuscript also aims to identify environmental factors that affect shark assemblages on sand banks, and to compare shark assemblages between undisturbed systems and sand banks where dredging is taking place.

The manuscript therefore sets out to accomplish a lot, though the long-term datasets available and the robust statistical treatment of the data on the author’s part allow for it.

The data is robustly collected, and the study set up allows the authors to make good inference about the seasonal and spatial trends of shark diversity and abundance at the study site. The combination of fishery-dependent and -independent methods strengthen the conclusions further. The statistical analysis is well planned and executed, lending credibility to the results. Sometimes the explanation of analytical techniques and decisions isn’t sufficiently clear/explicit, but can easily be made more accessible in review (see line-by-line comments below).

The results are mostly well presented, but the scope and variety of objectives sometimes results in parts being rushed and not sufficiently explained (see comments below). In particular, not enough attention is given to the comparison between the dredged and undisturbed site, despite this being one of the main aims of the manuscript. The role of the site as a nursery is also a key section in the discussion, but is not properly covered in the results. A better match between key points in the discussion and results sections would make this manuscript stronger.

The discussion is mostly well presented but, as suggested above, somewhat suffers from trying to achieve too much. The manuscript would probably benefit from a slightly more cohesive vision for the discussion. Some sections are not discussed in enough detail or aren’t supported by the results enough, while others could be avoided (see detailed comments below). The discussion would also perhaps benefit from a stronger conclusion. Nonetheless, the structure and language of the manuscript is clear and well thought-out, making for an interesting read.

Altogether, this is a well written manuscript and well executed study. The results presented are interesting and mostly well presented, supported by robust statistical analysis. Minor refinement are needed to hone-in its focus, but otherwise the manuscript is of very good quality.

Reviewer #2: The authors present a a thorough analysis of two large data sets to examine shark community composition at a large shoal complex. The results of their analysis are well presented and well explained, and they appear to have used appropriate statistical approaches to arrive at these results.

I did notice a few small typos here and there that should be corrected.

I have no reservations in recommending this manuscript for publication in this journal. Congratulations on a sound analysis and well written manuscript.

6. PLOS authors have the option to publish the peer review history of their article (what does this mean?). If published, this will include your full peer review and any attached files.

Reviewer #1: No

Reviewer #2: No

---

## [Author Response · Author response to Decision Letter 0]

5 May 2023

EDITOR COMMENTS

Dear Dr. Reyier,

Thank you for submitting your manuscript to PLOS ONE. After careful consideration, we feel that it has merit but does not fully meet PLOS ONE’s publication criteria as it currently stands. Therefore, we invite you to submit a revised version of the manuscript that addresses the points raised during the review process.

This is a strong and compelling manuscript, but four areas need to be strengthened before it can be published:

I. Introduction: Please re-phrase the goals of this study, to highlight that there is a hypothesis-driven component, with three objectives and an exploratory component. For instance, the introduction could explicitly state that the paper integrates two approaches towards understanding how the overall shark assemblage differs. The first hypothesis-driven approach addresses three patterns:

(1) spatially: comparing shallow shoals and adjacent deeper water

(2) temporally: comparing four seasons and

(3) compares the community at an active dredge site to a nearby control site.

The second exploratory component seeks to identify other habitat factors that help structure the shark community, and integrates a variety of bathymetric and hydrographic datasets. To strengthen this second component, I would ask the authors to explicitly list what patterns (or habitat variables) are being explored. 

Alternatively, if all of the environmental variables used to explore other habitat factors are already used in the hypothesis-driven approach, the paper could be focused on the first approach. Currently, its unclear how the exploratory component of the paper enhances the hypothesis-driven component. Are they really distinct? Is this disparity based on the multivariate analyses: the exploration part uses NMS ordinations and the hypothesis-driven part uses ANOVAs and PERMANOVAS?

Yes we understand, and one of the reviewers offered similar comments. We initially considered adding habitat covariates into the PERMANOVAs, thus merging the hypothesis-driven and exploratory aspects into global tests. In other words, do shark community differences exist across depth zones and seasons after accounting for temperature, turbidity, etc.? The guidance we’ve received from experts in multivariate community analyses is that these complex omnibus models can be difficult to interpret and that it’s often better to build the story by separating the two topics. 

With that in mind, we’ve revised our goals to better differentiate between the hypothesis-driven and exploratory components, and to name several of the important habitat covariates we considered in our exploratory aspect. We prefer to keep the community tests across depth zones and seasons grouped as a single goal. They were presented and modeled together in all instances, and hitting the reader with four separate goals might add confusion. Our goals statement now reads:

“The overarching purpose of this study is to document the shark community associated with a large sand shoal complex in east Florida, USA, a region of high shark diversity—and intensive research—that serves as an overwintering area for many species. Our specific goals are to explicitly test whether the shark assemblage differs between shallow shoals and adjacent deeper water, as well as across seasons, and to test whether the community observed at an active dredge site differs from that of a nearby undisturbed site. Moreover, we then explore how potentially influential habitat factors such as water depth, temperature, turbidity, and sediments, help explain patterns in the shark community.”

II. Methods: Sometimes the explanation of the analytical techniques and associated decisions are not sufficiently explicit but can easily be made more accessible in review (see line-by-line comments from one reviewer). One major area where improvements are needed is to provide detailed explanations about the environmental data and the way they were collected:

We struggled with how to strike a balance between detail and overall manuscript length. We’ve bulked up our Methods in several areas and hopefully addressed these concerns. These changes are described fully in the line-by-line responses below.

• Provide the depth range where the water column measurements were taken with the YSI sonde? Since there are shallow (< 6.1 m) and deep (6.1 – 20 m) stations, did the depth of the measurements vary widely? If so, can you please provide a summary of this information? 

Yes water quality was sampled mid-water at each site so values from sand shoal sites were taken at a shallower depth than offshore sites. We added mean depth at measurement for each stratum and inserted a reference to Table S1 to make the readers immediately aware that full summaries of all water quality and other habitat covariates are available if interested.

• Provide the make and model of the YSI sonde and the resolution of the physical measurements. Moreover, please provide information about any validation of the data and about any blanks taken in the field or in the lab before/ after the field measurements. For instance, using type-II water.

We added the make/model of the sonde and that it was “calibrated daily”. In general, the current generation of water quality sondes are very accurate and most fisheries surveys like ours don’t provide much detail on the calibration process. We can bulk up these details here if required though. The supplemental tables also provide the unit resolution.

• Provide details about how the water depth and seafloor slope were collected. Was the depth measured at the start and at the end of the set, and then the gradient was calculated as max depth minus min depth divided by the linear distance between them? 

We just used depth range for each set as our slope, calculated as the absolute value of the difference between the start and end depth on each set (measured by the boat’s depth sounder). Since all longline sets spanned the same distance, we considered this sufficient since there is no need to standardize. We’ve added language to clarify.

• Many variables are discussed, but there is no explanation of how they were collected, what is the resolution and precision / accuracy of the measurements, and how were these data merged with the acoustic and longline catch data. Were these discrete data interpolated to assign values to the shark data? I strongly suggest the authors provide a summary table, showing these details for all the environmental data.

• 

We have now added considerable detail regarding individual habitat variables (with our edits aided by reviewer inputs below). We also made a structural change in the manuscript by moving the longline and telemetry habitat collection paragraphs closer together in the Methods. Hopefully this improves readability.

We also offer summaries (units, means, ranges, precision) of longline and acoustic telemetry variables as a supplemental Tables S1 and S2. Our thinking was that we already present three tables in the main manuscript body and that adding more could be cumbersome for the reader. We also inserted references to these supplemental tables earlier in the Methods so the reader knows these habitat details are accessible.

• In particular, please provide more details about the temporal / spatial resolution of the remotely sensed data? Also, please explain how were clouds dealt with? Did you use or develop seasonal composites? Did you use the maximum or mean values to do so? 

We’ve added the spatial and temporal resolution of both remotely sensed temperature and turbidity (Kd490) datasets and confirmed that we developed monthly averages for each station. Clouds are an inherent issue (one reason we went with an 8-day composite for turbidity) but in these queries, when clouds obscure readings, null values are returned and do not weigh heavily on monthly means. In other words, these data are quality controlled before they are even made available to the public. As an aside, we compared these SST values to local buoys and some in-water temperature loggers we deployed and were very happy with how closely they matched.

• It seems strange that the definition of the offshore domains is different for shoals and reefs: offshore shoal stations (>1.5–13 km from shore) VERSUS offshore reef stations (10–22 km from the shore). Why not do two offshore sections (shoal and reef) up to 13 km from shore and then consider a very offshore region (13 – 22 km from shore). I am concerned that the current offshore stations are sampling different ocean habitats. Can you show with your environmental data that they do not differ in depth and water properties?

This was confusing as first written. Our shoals are inshore of the reef tract in our region. There isn’t a way to monitor shoal and reef habitats with similar depth and distance from shore characteristics. Our rationale is that, in addition to a dredge and control comparison, we wanted to compare nearshore shoal stations vs. offshore shoal stations since we suspect some species use the coastline for navigation (the Sequeira 2018 paper we cite provides a rationale for this). We also wanted to examine reef stations since we know some sharks (e.g., sandbar, tiger, nurse) are strong reef associates in east Florida. These factors could (and did) lead to somewhat differing assemblages. 

The distance from shore values we listed were the actual range of deployment distance of our receivers, not predefined zones. We also used ‘offshore’ to describe both shoals and reefs when we actually do want to consider them different habitat classifications. Both choices were confusing. To address this, the text now reads:

“Separately, a simple one-way PERMANOVA on telemetry data was used to test for assemblage differences between five more specific receiver station groups. These included the dredge site and control site, as well as all other nearshore shoal stations (<1.5 km from shore), offshore shoal stations (>1.5 from shore), and reef stations. The rationale for these latter three groups is that some sharks may closely follow the shoreline for navigation while others are known reef associates, behaviors which may also influence community patterns within the study area.” 

• Can you explain why a 2-D solution was selected for the MDS ordination? Was a scree plot used? Were the 2-D solutions the ones with the lowest stress? Or were 2-D solutions selected for the ease of visual interpretation? 

Our choice was just based on written guidance from the package developers that stress > 0.2 in nMDS plots can be misleading should not be presented in 2D. We were well under this threshold in each case. Moreover, the software simultaneously generates 2D and 3D plots and we found major sample groupings were visible and consistent regardless of perspective. I personally find 3D plots cumbersome to present so avoid them unless truly needed. 

• Regarding the comparison of the dredge and the control site, you stated that: “Combined across the six-year study, a similar number of species (15 vs. 16) and individuals (314 vs. 309) were detected by acoustic receivers at the sand dredge site vs. nearby control site (Table 3).” Could you generate rarefaction curves to compare the species richness and the diversity (and evenness) at these two sites? This would be an easy way to visualize these patterns? You could use the iNext R package.

We liked this idea and have added a rarefaction plot that estimates the relationship between species richness and count of tagged sharks. We actually did this for all five of our habitat groupings and added it as a separate panel of Fig. 7 (whose other half is an MDS ordination of the same five groups). We used PRIMER’s rarefaction tool, mostly because I’m most familiar with it. I haven’t seen rarefaction curves used for diversity indices or evenness before. We also added supporting text in the Methods and Results.

III) Discussion: A more focused and cohesive discussion, built upon the results, would strengthen the ms. 

• The broad scope of the objectives sometimes results in parts of the discussion not being sufficiently explained (see reviewer comments). For instance, not enough attention is given to the comparison between the dredged and undisturbed site, despite this being one of the main aims of the manuscript. 

• The manuscript would also benefit from a slightly more cohesive discussion, built upon the results. Some sections are not discussed in enough detail or are not well supported by the results, while others could be avoided (see detailed comments from reviewer). Namely, the role of the dredge site as a nursery is a key section in the discussion but is not properly covered in the results.

• The discussion would also benefit from a stronger conclusion. Perhaps this could be enhanced by streamlining the goals listed in the introduction and explicitly addressing these points.

No arguments here. We’ve added additional language (and caveats) regarding dredge and control comparisons, refined our nursery discussion (plus more results earlier in the ms), clarified our longline vs. telemetry comparison, and added a dedicated Conclusions paragraph so the ms doesn’t end on language regarding methods. The responses below offer details.

One important note. We don’t want to overemphasize our dredge vs. control comparison. Our dredge site has been exploited for sand several times already so our assembles comparison cannot provide any causal linkages to dredging. We’ve alluded to this in the Discussion by stating:

“One major limitation of the present study is that the shark community comparison at the dredge vs. control site was purely descriptive and complicated by the repeated disturbance of the dredge site over the past 20 years. More rigorous study designs such as Before-After-Control-Impact (BACI) protocols are viable options when evaluating proposed sand borrow sites and would better reveal causal links between dredging to shark community structure.”

IV) Finally, a few typos should be corrected. Please read the paper carefully before re-submitting.

Done. Hopefully found them all.

We look forward to receiving your revised manuscript.

Kind regards,

David Hyrenbach, Ph.D.

Academic Editor

PLOS ONE

REVIEWER COMMENTS

1. Is the manuscript technically sound, and do the data support the conclusions? 

Reviewer #1: Partly

Reviewer #2: Yes

2. Has the statistical analysis been performed appropriately and rigorously? 

Reviewer #1: Yes

Reviewer #2: Yes

3. Have the authors made all data underlying the findings in their manuscript fully available? 

Reviewer #1: Yes

Reviewer #2: Yes

4. Is the manuscript presented in an intelligible fashion and written in standard English? 

Reviewer #1: Yes

Reviewer #2: Yes

Other comments: 

This manuscript makes use of long-term longline catch data and passive acoustic telemetry to describe the shark assemblages of a sand bank system and investigate the temporal and spatial patterns in their distribution and abundance. The manuscript also aims to identify environmental factors that affect shark assemblages on sand banks, and to compare shark assemblages between undisturbed systems and sand banks where dredging is taking place. The manuscript therefore sets out to accomplish a lot, though the long-term datasets available and the robust statistical treatment of the data on the author’s part allow for it.

The data is robustly collected, and the study set up allows the authors to make good inference about the seasonal and spatial trends of shark diversity and abundance at the study site. The combination of fishery-dependent and -independent methods strengthen the conclusions further. The statistical analysis is well planned and executed, lending credibility to the results. Sometimes the explanation of analytical techniques and decisions isn’t sufficiently clear/explicit, but can easily be made more accessible in review (see line-by-line comments below). 

The results are mostly well presented, but the scope and variety of objectives sometimes results in parts being rushed and not sufficiently explained (see comments below). In particular, not enough attention is given to the comparison between the dredged and undisturbed site, despite this being one of the main aims of the manuscript. The role of the site as a nursery is also a key section in the discussion, but is not properly covered in the results. A better match between key points in the discussion and results sections would make this manuscript stronger. 

The discussion is mostly well presented but, as suggested above, somewhat suffers from trying to achieve too much. The manuscript would probably benefit from a slightly more cohesive vision for the discussion. Some sections are not discussed in enough detail or aren’t supported by the results enough, while others could be avoided (see detailed comments below). The discussion would also perhaps benefit from a stronger conclusion. Nonetheless, the structure and language of the manuscript is clear and well thought-out, making for an interesting read. 

Altogether, this is a well written manuscript and well executed study. The results presented are interesting and mostly well presented, supported by robust statistical analysis. Minor refinement are needed to hone-in its focus, but otherwise the manuscript is of very good quality. 

Line by line comments: 

45 – References here could contain the more recent Dulvy lab papers 

Sure. We’ve added Pacourea et al. (2021) and Sherman et al. (2023), both from this lab.

48 – Instead of “most sharks”, specify if you mean shark species or populations 

Changed to “most species now have recognized Essential Fish Habitat…”

60 – this sentence needs a little more explanation of what the authors mean by “a unique perspective on shark community dynamics”, and perhaps some supporting references. 

See response below

60 – 63 This line of thought is a little rushed and could use further explanation 

Addressing both comments above. I revised this sentence to:

“Year-round sampling of sharks associated with specific geomorphological features or substrates of the continental shelf are uncommon but can provide details on seasonal abundance trends and habitat associations that are often lacking in coarse-scale regional surveys.”

I am having a hard time finding a good references for this statement. Multi-year, year-round shark surveys over the open shelf truly are uncommon. 

67 – missing references (after “ when compared to adjacent deeper water”) 

Added Michel et al. (2013)

87 – While the introduction flows well, at this point I find myself wondering why these questions are of “central importance”. Perhaps a little more detail on the ecological relevance of the aims of the paper and their role in explaining impacts from dredging would help in making this section stronger. 

See response to Editor above regarding our goals statements. 

We are somewhat hesitant to focus too intently on the dredging aspect of this work. Our dredge site has been repeatedly disturbed starting 10 years before our study started. Our comparisons are purely observational in nature, and with no ability to conduct a more rigorous BACI-type design. We’ve added important language in this discussion to allude to this shortcoming.

90 – 94 The comparison of methodologies feels more appropriate for the discussion 

We would lobby to retain this sentence in some form. Pairing traditional fisheries-independent sampling and electronic tagging is still uncommon in shark community surveys and is a unique selling point of our study. An edit was added to convey that this paired approach ‘strengthened’ our findings but we can drop the sentence at the editor’s discretion.

135 – Could be good to give a summary statistic for soak times (mean and standard deviation) 

Good idea. Added mean and sd values. Our actual soak time (48 min) was longer than our minimum planned soak time (30 min) due to the time spent processing and tagging animals. But catch was standardized to CPUE, and raw time values will be present in data supplement if readers need more detail.

191 – how were mortality events identified? 

Basically tags that didn’t move. Changed language to:

 “All shark detections were then combined into a single database and screened with a custom R script to remove any non-moving tags (presumed mortality events) or false detections….”

202 – How was “shark community” quantified/measured? 

See response below

203 – It is quite difficult to understand what averaging means in this context – what quantity exactly was averaged? 

See response below

205 – Would be good to make clearer that study year was not considered a factor ONLY in acoustic data analysis. Sometimes explanation of analytical details for the two methods are hard to distinguish and separate throughout the methods section.

Addressing all three comments above. We agree this description of sample averaging was hazy and have revised with more detail. These edits were not trivial and the ‘track changes’ version shows them best.

240 – Again, what do the authors mean by averaged here? More detail is necessary 

Agreed. Our revised and more detailed description of sample averaging earlier in the methods should improve clarity here as well. Therefore we didn’t make changes to this particular line.

249 – Perhaps a brief explanation of sediment percent organics and percent fine estimates is needed here for audiences who are not familiar with these quantities.

Sure. Added additional detail.

267 – The authors should make it clear that the difference between deep and shallow areas is not apparent when sharpnose shark catches are excluded, and that there is much inter-specific variation in patterns of abundance between depths. 

We did not make any edits here. In lines 271-273 of the original ms, we already state:

“When sharpnose were excluded from this comparison, shark CPUE was modestly but significantly greater in shallow water (4.0 vs 3.1 sharks per 100 hook-hours; p = 0.031) but did not differ across seasons (p = 0.387).” 

We’re assuming the reviewer just missed this subsequent statement.

275 – Missing reference to Figure 3 

Added

278 – Could the female-skewed sex ratios for cool season sharks have something to do with the existence of nurseries? Perhaps female sharks coming in to pup at this time? This perhaps warrants further attention in results to have stronger ties to the discussion. 

This is a logical assumption based on our text but this sex bias peaks in winter 3-4 months prior to spring pupping periods. To convey the time of year where this phenomenon was most apparent, and to add more context regarding nursery value, we’ve revised to:

 “This female-skewed distribution was most apparent from December through March, well before the typical spring pupping period, although gravid blacknose (n = 20), blacktip (n = 12), and finetooth (n = 26) were encountered over a wider timeframe from January to April or May each year”

The section is also missing a better description of length measurement distributions for the different species, which again would provide a better link to the section on nurseries in the discussion. 

For the sake of brevity, we do not present complete length distributions for each species since these generally consist of multiple single-species histograms. That said, we have provided a brief narrative of shark size classes earlier in this section and provided length means and ranges in Table 2. We will make the full longline dataset available as a supplemental and have revised our discussion of shark nurseries.

281 – The CPUE comparisons are not clear, and the lowercase codes are not explained anywhere so they are impossible to understand. This figure caption needs much more detail to make the figure legible. 

Agreed. Boxplots summarize several aspects of a dataset and while they have fairly standard definitions, the definitions aren’t intuitive from the graphics alone (I remember having to recently review them myself). We’ve changed caption from: 

“Fig. 2. Shark longline CPUE (sharks per 100 hook-hours) averaged by water depth zone and season for (A) all species, and (B) excluding Atlantic sharpnose shark. Pairwise CPUE difference across seasons (significant only for the all-species test) are denoted with lowercase letters” 

to

“Boxplots of shark longline CPUE across water depth zones and seasons for (A) all species, and (B) excluding Atlantic sharpnose shark. Horizontal black lines represent the group median, boxes represent the interquartile range, and whiskers represent the full data range. Seasonal differences were only detected for the all-species ANOVA test with pairwise differences denoted with lowercase letters.”

Also, to improve readability, we changed the whiskers to encompass outliers in the data range. Outliers often denote suspect data but were just atypical samples that caught LOTS of sharks. 

293 – 298 This feels a bit more fit for the discussion rather than the results section 

We’ve removed the phrase “…reflecting the current high interest in tagging studies of large coastal sharks” which is admittedly a statement not based on our results. 

344 – change reference to fig 6 with “(Fig. 6 A and B, respectively)” 

Done

352 – I think section would benefit from a deeper analysis of the results. For example, the shark communities at dredged and control sites look separate in figure 7, but this isn’t touched on in the section. It would also be good to look into residency times at the two sites and report any potential differences. If dredging activities took place during the course of the study, another interesting metric would also be the time between dredging taking place and the first detection of sharks at the site. As the effects (or lack thereof) of dredging sandbanks on shark communities is one of the main foci of the paper, this section really needs a bit more attention. 

For the dredge vs. control comparison, we’ve bolstered our rationale for the five habitat classifications in the Methods (now lines 250-255) and added rarefaction curves in the Results (now Fig. 7A) to compare patterns in species richness while accounting for uneven numbers of monitoring stations in each habitat. We also now state that:

“Nonetheless, a PERMANOVA test and follow-on pairwise comparisons confirmed subtle but significant community differences between the dredge and control site as well as other receiver groups…”

Regarding residency and direct dredging impacts, we totally agree and have explored both topics in our work at Cape Canaveral. We did not present our findings here because they are largely single-species behavioral analyses. We felt that it detracted from the community focus of the paper and would greatly expand manuscript length and complexity. 

That said, in the Discussion (now lines 569-572), we’ve added a summary and reference to these findings that are available in our agency report.

411 – perhaps, given the very narrow depth range, you could refer to “habitat differences” (ridge vs adjacent waters) rather than “shallow vs deep”? 

The original sentence (below) provides more detail than just “shallow vs deep” so we opted not to make any edits here.

“Longlines and acoustic telemetry independently documented real but modest differences in the shark community between shallow shoal ridges and deeper water along shoal flanks and troughs”

451 – Change “due to “ with “thanks to” 

Done

504 – As mentioned above, this section seems, at the moment, a little disconnected from the aims of the paper and the results presented. As it is of interest to the ecology of this habitat, I would suggest that the authors make more explicit reference to it in the introduction and better highlight figures/numbers that pertain to this in the results. For example, the numbers in lines 514-517 are only found in the table but never specifically highlighted in the text. This would also be a good place to reflect on possible links to the highly female-skewed sex ratios registered in cold seasons. 

We’ve made lots of edits in the section on shark nursery values including moving the numbers of neonates captured from the Discussion to the Results. We’ve also added language in the Discussion (now lines 552-554) stating that pupping is likely occurring north of our study site. 

517 – change “underestimating” with “underestimated”. 

Done

521 – can you give any info/detail on the sizes or estimated ages of these sharks? 

Yes we actually have excellent size range estimates of these animals based on previous sampling. We updated sentence (now line 548) to read “nearly 1800 immature (~0.75-2.0 m TL) lemon sharks”. We don’t have good size-age relationships for Florida yet though.

526 – The conclusion to this section would be stronger if it focused on ecological/conservation importance of the findings, rather than methodological considerations.

Agreed. We’ve added a dedicated Conclusion that ends with a note on management

527 – This section feels important but contains conjectures and/or claims that aren’t totally supported by the results at the moment. The data would however allow for a deeper analysis of differences (composition, residency, seasonality) that really is needed to support this section (see some comments above). 

We didn’t want delve into single species behavior in this ms but have added the following sentence:

“Moreover, behavioral assessments of the five shark species tagged at Cape Canaveral (detailed in [27]) demonstrated that individuals remained near a given receiver station for less than an hour on average, made regular inshore-offshore movements, and commonly undertook migrations spanning several hundred kilometers, all behaviors which minimize the effects of localized dredging disturbances”

557 – it is not really the shark species itself that is designated as an “Habitat Area of Particular concern”. Perhaps it would be best to rephrase the sentence slightly to something like “one of only three shark species in the US to benefit from these designations”. Also, Specify you are referring to BEACH nourishment. 

Changes made.

563 – This section feels a little rushed and perhaps doesn’t add as much to the manuscript as other sections do/could. I would suggest that the authors consider removing this section in favour of giving more attention to the sections on nurseries and effects of dredging. This would hone the focus of the manuscript on ecology and conservation and make the work stronger. 

We’re hoping to retain a brief discussion as to the relative merits of longlines and acoustic telemetry since they historically haven’t been paired together for community assessments but likely will be in the coming years. We feel this section can provide some valuable insights to future researchers. We’ve instead reduced and hopefully strengthened this section to meet these reviewer concerns.

The section may also seem out of place because it was found at the end of the manuscript and wasn’t a strong concluding topic. We’ve now added a dedicated Conclusion paragraph that highlight only ecological and management topics.

587 – The manuscript presents many interesting results on ecology and conservation of sharks in this habitat and geographic area. It is therefore a shame to conclude on technicality of methodologies that are perhaps more obvious. I suggest the authors sacrifice the section above in favour of a stronger conclusion that brings together the results of the study and perhaps offers recommendation on the management of sandbanks for shark conservation. 

Agreed. See comment above.

Fig 1 – Perhaps the dots in A could be changed to make apparent which sets belong to each year of the study? In B, I suggest the authors make it clearer that the circle of dots further north is the control site. 

We’ve added a label to identify the control site, and this comment also made us realize we excluded the labels for the shoals themselves and other important features. A good catch! 

We would prefer to not color-code longline sets by calendar year. Sampling occurred over six years (2012-17). Color coding will add extra complexity to the map and legend, and since sites were randomly selected, no new patterns would be revealed. 

Fig 2. As mentioned above, I cannot quite understand what the letters above the boxplots are trying to tell me. Also, did you consider varying the width of the boxes by the sample size, or using violin/rainclouds plots to better represent the underlying distribution of the data? The simplicity of boxplots sometimes can mask subtle differences that become apparent when using these other visualisation tools. 

See comment above re: boxplots. We’ve expanded the caption to (hopefully) improve understanding and simplified the plot by removing outliers (which aren’t really outliers, just very ‘sharky’ samples). Varying the width of the boxes won’t help here because sample size was by design held standard across depths and seasons. All plots within each panel would still be the same width.

Rainfall plots LOOK COOL but would be messy here since these graphs are drawn from 900 longline sets. We are very open to violin plots instead if the editor prefers. Our thought is that simple is best in this instance.

Fig 3. I really like the level of detail and ease of comparison offered by the figure. Would it be possible to also add a figure (though I realise there are many already) showing maps of species distributions in the area? 

We offered some of these in the final project report (see pg. 58-61 here for examples). They just eat up a ton of space and we didn’t want to dwell too much on single species products. We could reference this report again if needed. 

Fig 6. This is a minor comment, but in all other figure you have used these shades of blue to indicate different depths, so it took me a while to realise the colours in this figure actually indicated different sampling strategies. Perhaps the colours could be changed to make the distinction more immediate? The same also applies to figure 8. 

This is a good point. Colors now changed to red and blue so they don’t imply water depth.

Fig 7. In caption, the authors use “borrow” instead of “dredging”. I suggest keeping to the latter for consistency with the rest of the manuscript.

Nice catch. Change made.

---

## [Editor Report · Decision Letter 1]

22 May 2023

Sharks associated with a large sand shoal complex: Community insights from longline and acoustic telemetry surveys

PONE-D-23-02711R1

Dear Dr. Reyier,

We’re pleased to inform you that your manuscript has been judged scientifically suitable for publication and will be formally accepted for publication once it meets all outstanding technical requirements. We thank you for addressing all the reviewer comments and providing a clear and compelling rationale for your revisions. 

Kind regards,

David Hyrenbach, Ph.D.

Academic Editor

PLOS ONE

---

## [Editor Report · Acceptance letter]

9 Jun 2023

PONE-D-23-02711R1 

Sharks associated with a large sand shoal complex: Community insights from longline and acoustic telemetry surveys 

Dear Dr. Reyier:

I'm pleased to inform you that your manuscript has been deemed suitable for publication in PLOS ONE. Congratulations! Your manuscript is now with our production department. 

Kind regards, 

on behalf of

Dr. David Hyrenbach 

Academic Editor

PLOS ONE